# HOW DNNs BREAK THE CURSE OF DIMENSIONALITY: COMPOSITIONALITY AND SYMMETRY LEARNING

**Arthur Jacot, Seok Hoan Choi & Yuxiao Wen**
Courant Institute
New York University
New York, NY 10012
{arthur.jacot,shc443,yuxiaowen}@nyu.edu

## ABSTRACT

We show that deep neural networks (DNNs) can efficiently learn any composition of functions with bounded $F_1$-norm, which allows DNNs to break the curse of dimensionality in ways that shallow networks cannot. More specifically, we derive a generalization bound that combines a covering number argument for compositionality, and the $F_1$-norm (or the related Barron norm) for large width adaptivity. We show that the global minimizer of the regularized loss of DNNs can fit for example the composition of two functions $f^* = h \circ g$ from a small number of observations, assuming $g$ is smooth/regular and reduces the dimensionality (e.g. $g$ could be the quotient map of the symmetries of $f^*$), so that $h$ can be learned in spite of its low regularity. The measures of regularity we consider is the Sobolev norm with different levels of differentiability, which is well adapted to the $F_1$ norm. We compute scaling laws empirically and observe phase transitions depending on whether $g$ or $h$ is harder to learn, as predicted by our theory.

## 1 INTRODUCTION

One of the fundamental features of DNNs is their ability to generalize even when the number of neurons (and of parameters) is so large that the network could fit almost any function (Zhang et al., 2017). Actually DNNs have been observed to generalize best when the number of neurons is infinite (Belkin et al., 2019; Geiger et al., 2019; 2020). The now quite generally accepted explanation to this phenomenon is that DNNs have an implicit bias coming from the training dynamic where properties of the training algorithm lead to networks that generalize well. This implicit bias is quite well understood in shallow networks (Chizat & Bach, 2018; Rotskoff & Vanden-Eijnden, 2018), in linear networks (Gunasekar et al., 2018; Li et al., 2020), or in the NTK regime (Jacot et al., 2018), but it remains ill-understood in the general deep nonlinear case.

In both shallow networks and linear networks, one observes a bias towards small parameter norm (either implicit (Chizat & Bach, 2020) or explicit in the presence of weight decay (Wang & Jacot, 2024)). Thanks to tools such as the $F_1$-norm (Bach, 2017), or the related Barron norm (Weinan et al., 2019), or more generally the representation cost (Dai et al., 2021), it is possible to describe the family of functions that can be represented by shallow networks or linear networks with a finite parameter norm. This was then leveraged to prove uniform generalization bounds (based on Rademacher complexity) over these sets (Bach, 2017), which depend only on the parameter norm, not on the number of neurons or parameters.

Similar bounds have been proposed for DNNs (Bartlett et al., 2017; Golowich et al., 2020; Barron & Klusowski, 2019; Schmidt-Hieber, 2020; Nitanda & Suzuki, 2020; Hsu et al., 2021; Sellke, 2024), relying on different types of norms on the parameters of the network. Analogues of these bounds have been extended to other architectures, such as CNNs (Ledent et al., 2021; Graf et al., 2022; Truong, 2022; Galanti et al., 2023). But there remains many issues: these bounds are typically orders of magnitude too large (Jiang et al., 2019; Gonon et al., 2023), and they tend to explode as the depth $L$ grows (Sellke, 2024).

It is difficult to compare these bounds, since they involve many terms that have complex behavior as the depth increases. To justify our new complexity measure $R(\theta)$ and bound, we pair it with an

approximation result on composition of Sobolev functions (functions with a certain decay of their Fourier or spherical harmonics coefficients), which implies that DNNs can learn such composite functions with almost optimal rates, allowing to break the curse of dimensionality in certain cases.

Previous works has shown that DNNs with either bounded width/depth (Schmidt-Hieber, 2020) or with bounded number of non-zero parameters (Galanti et al., 2023; Poggio & Fraser, 2024) can effectively learn such compositional functions. This paper shows that these strong sparsity assumptions are not strictly necessary for DNNs to learn such compositional functions.

The family of composite Sobolev functions is also a useful theoretical baseline to compare different norm-based bounds, by checking which complexity measure lead to tighter rates. This allows us to show the importance of replacing operator norms by Lipschitz constants and other design choices.

## 1.1 CONTRIBUTION

We consider Accordion Networks (AccNets), which are the composition $f_{L:1} = f_L \circ \cdots \circ f_1$ of multiple shallow networks $f_\ell(x) = W_\ell \sigma(V_\ell x + b_\ell)$, we prove a uniform generalization bound for the MSE $\sqrt{\mathcal{L}(f_{L:1})} - \sqrt{\tilde{\mathcal{L}}_N(f_{L:1})} \lesssim \sqrt{\frac{R(\theta)}{\sqrt{N}}}$, for a complexity measure

$$R(\theta) = \left( \prod_{\ell=1}^{L} Lip(f_\ell) \right) \left( \sum_{\ell=1}^{L} \frac{\|f_\ell\|_{F_1}}{Lip(f_\ell)} \sqrt{d_\ell + d_{\ell-1}} \right)$$

that depends on the $F_1$-norms $\|f_\ell\|_{F_1}$ (which is upper bounded by the parameter norm of the corresponding shallow network $\|W_\ell\|_F^2 + \|V_\ell\|_F^2 + \|b_\ell\|^2$) and Lipschitz constant $Lip(f_\ell)$ of the subnetworks, and the intermediate dimensions $d_0, \ldots, d_L$. This use of the $F_1$-norms makes this bound independent of the widths $w_1, \ldots, w_L$ of the subnetworks, though it does depend on the depth $L$ (it typically grows linearly in $L$ which is still better than the exponential growth often observed).

Any traditional DNN can be mapped to an AccNet (and vice versa), by spliting the middle weight matrices $W_\ell$ with SVD $USV^T$ into two matrices $U\sqrt{S}$ and $\sqrt{S}V^T$ to obtain an AccNet with dimensions $d_\ell = \text{Rank}W_\ell$, so that the bound can be applied to traditional DNNs with bounded rank.

We then show an approximation result: any composition of Sobolev functions $f^* = f_{L^*}^* \circ \cdots \circ f_1^*$ can be approximated with a network with either a bounded complexity $R(\theta)$ or a slowly growing one. Thus under certain assumptions one can show that DNNs can learn general compositions of Sobolev functions (i.e. functions whose first $\nu \in \mathbb{N}$ derivatives are bounded in $L_2$ norm). This ability can be interpreted as DNNs being able to learn symmetries, allowing them to avoid the curse of dimensionality in settings where kernel methods or even shallow networks suffer heavily from it.

Empirically, we observe a good match between the scaling laws of learning and our theory, as well as qualitative features such as transitions between regimes depending on whether it is harder to learn the symmetries of a task, or to learn the task given its symmetries.

## 2 ACCORDION NEURAL NETWORKS AND RESNETS

Our analysis is most natural for a slight variation on the traditional fully-connected neural networks (FCNNs), which we call Accordion Networks, which we define here. Nevertheless, all of our results can easily be adapted to FCNNs.

Accordion Networks (AccNets) are simply the composition of $L$ shallow networks, that is $f_{L:1} = f_L \circ \cdots \circ f_1$ where $f_\ell(z) = W_\ell \sigma(V_\ell z + b_\ell)$ for the nonlinearity $\sigma : \mathbb{R} \to \mathbb{R}$, the $d_\ell \times w_\ell$ matrix $W_\ell$, $w_\ell \times d_{\ell-1}$ matrix $V_\ell$, and $w_\ell$-dim. vector $b_\ell$, and for the widths $w_1, \ldots, w_L$ and dimensions $d_0, \ldots, d_L$. We will focus on the ReLU $\sigma(x) = \max\{0, x\}$ for the nonlinearity. The parameters $\theta$ are made up of the concatenation of all $(W_\ell, V_\ell, b_\ell)$. More generally, we denote $f_{\ell_2:\ell_1} = f_{\ell_2} \circ \cdots \circ f_{\ell_1}$ for any $1 \leq \ell_1 \leq \ell_2 \leq L$.

We will typically be interested in settings where the widths $w_\ell$ is large (or even infinitely large), while the dimensions $d_\ell$ remain finite or much smaller in comparison, hence the name accordion.

If we add residual connections, i.e. $f_{L:1}^{res} = (f_L + id) \circ \cdots \circ (f_1 + id)$ for the same shallow nets $f_1, \ldots, f_L$ we recover the typical ResNets.

*Remark.* The only difference between AccNets and FCNNs is that each weight matrix $M_\ell$ of the FCNN is replaced by a product of two matrices $M_\ell = V_\ell W_{\ell-1}$ in the middle of the network (such a structure has already been proposed (Ongie & Willett, 2022; Parkinson et al., 2023)). Given an AccNet one can recover an equivalent FCNN by choosing $M_\ell = V_\ell W_{\ell-1}$, $M_0 = V_0$ and $M_{L+1} = W_L$. In the other direction there could be multiple ways to split $M_\ell$ into the product of two matrices, but we will focus on taking $V_\ell = U\sqrt{S}$ and $W_{\ell-1} = \sqrt{S}V^T$ for the SVD decomposition $M_\ell = USV^T$, along with the choice $d_\ell = \text{Rank}M_\ell$. One can thus think of AccNets as rank-constrained FCNNs.

## 2.1 LEARNING SETUP

We consider a traditional MSE regression setup, where we want to find a function $f : \Omega \subset \mathbb{R}^{d_{in}} \to \mathbb{R}^{d_{out}}$ that minimizes the population loss $\mathcal{L}(f) = \mathbb{E}_{x\sim\pi} \|f(x) - f^*(x)\|^2$ for the 'true function' $f^*$, and input distribution $\pi$. Given a training set $x_1, \ldots, x_N$ of size $N$ we approximate the population loss by the empirical loss $\tilde{\mathcal{L}}_N(f) = \frac{1}{N}\sum_{i=1}^N \|f(x_i) - f^*(x_i)\|^2$ that can be minimized.

To ensure that the empirical loss remains representative of the population loss, we will prove high probability bounds on the generalization gap $\sqrt{\mathcal{L}(f)} - \sqrt{\tilde{\mathcal{L}}_N(f)} = O(\sqrt{R}N^{-\frac{1}{2}})$ uniformly over function families with complexity bounded by $R$. Thus as long as the train error is small enough $\tilde{\mathcal{L}}_N(f) = O(RN^{-\frac{1}{2}})$, so will the test error $\mathcal{L}(f) = O(RN^{-\frac{1}{2}})$. This type of bound on the gap between squared roots of the losses is more natural for the MSE loss and allows for so-called fast rates under the assumption that the train error is itself small. Note that our generalization bounds rely on covering number arguments, so they could easily be applied to Lipschitz losses using Dudley's theorem.

For simplicity of analysis, we will assume that the true function $f^*$ and estimator $\hat{f}$ are uniformly bounded $\|f^*\|_\infty, \|\hat{f}\|_\infty \le B$. This can be guaranteed easily by adding a renormalizing nonlinearity at the end of the network $\gamma_B(x) = \frac{Bx}{\max\{B, \|x\|\}}$. Since this is a contraction it does not affect the covering number argument that we rely on.

## 3 GENERALIZATION BOUND FOR DNNS

The reason we focus on accordion networks is that there exists generalization bounds for shallow networks (Bach, 2017; Weinan et al., 2019), that are (to our knowledge) widely considered to be tight, which is in contrast to the deep case, where many bounds exist but no clear optimal bound has been identified. Our strategy is to extend the results for shallow nets to the composition of multiple shallow nets, i.e. AccNets. Roughly speaking, we will show that the complexity of an AccNet $f_\theta$ is bounded by the sum of the complexities of the shallow nets $f_1, \ldots, f_L$ it is made of.

We will therefore first review (and slightly adapt) the existing generalization bounds for shallow networks in terms of their so-called $F_1$-norm (Bach, 2017), and then prove a generalization bound for deep AccNets.

## 3.1 SHALLOW NETWORKS

The complexity of a shallow net $f(x) = W\sigma(Vx + b)$, with weights $V \in \mathbb{R}^{w\times d_{in}}$ and $W \in \mathbb{R}^{d_{out}\times w}$, can be bounded in terms of the quantity $C = \sum_{i=1}^w \|W_{\cdot i}\|\sqrt{\|V_{i\cdot}\|^2 + b_i^2}$. First note that the rescaled function $\frac{1}{C}f$ can be written as a convex combination $\frac{1}{C}f(x) = \sum_{i=1}^w \frac{\|W_{\cdot i}\|\sqrt{\|V_{i\cdot}\|^2 + b_i^2}}{C}\bar{W}_{\cdot i}\sigma(\bar{V}_{i\cdot}x + \bar{b}_i)$ for $\bar{W}_{\cdot i} = \frac{W_{\cdot i}}{\|W_{\cdot i}\|}$, $\bar{V}_{i\cdot} = \frac{V_{i\cdot}}{\sqrt{\|V_{i\cdot}\|^2 + b_i^2}}$, and $\bar{b}_i = \frac{b_i}{\sqrt{\|V_{i\cdot}\|^2 + b_i^2}}$, since the coefficients $\frac{\|W_{\cdot i}\|\sqrt{\|V_{i\cdot}\|^2 + b_i^2}}{C}$ are positive and sum up to 1. Thus $f$ belongs to $C$ times the convex hull

$$B_{F_1} = \text{Conv}\left\{x \mapsto w\sigma(v^Tx + b) : \|w\|^2 = \|v\|^2 + b^2 = 1\right\}.$$

This set can be thought as the unit ball w.r.t. to the $F_1$-norm (Bach, 2017). The $F_1$-norm $\|f\|_{F_1}$ itself can then be defined as the smallest positive scalar $s$ such that $\frac{1}{s}f \in B_{F_1}$. Note that by the AM-GM

inequality, we have

$$\|f\|_{F_1} \leq C = \sum_{i=1}^{w} \|W_{\cdot i}\| \sqrt{\|V_{i\cdot}\|^2 + b_i^2} \leq \frac{1}{2} \sum_{i=1}^{w} \|W_{\cdot i}\|^2 + \|V_{i\cdot}\|^2 + b_i^2$$

The generalization gap over any $F_1$-ball can be uniformly bounded with high probability:

**Theorem 1.** *For any input distribution $\pi$ supported on the $L_2$ ball $B(0, b)$ with radius $b$, we have with probability $1 - p$, over the training samples $x_1, \ldots, x_N$, that for all $f \in \{f : \|f\|_{F_1} \leq R, \|f\|_\infty \leq B\}$*

$$\sqrt{\mathcal{L}(f)} - \sqrt{\tilde{\mathcal{L}}_N(f)} \leq c_0 \sqrt{BRN^{-\frac{1}{2}}} + c_1 B \sqrt{\frac{-\log p/2}{N}}.$$

*Therefore if $\tilde{\mathcal{L}}_N(f) = O(BRN^{-\frac{1}{2}})$ then $\mathcal{L}(f) = O(BRN^{-\frac{1}{2}})$.*

This theorem is a slight variation of the one found in (Bach, 2017): we simply generalize it to multiple outputs, and apply to the (non-Lipschitz) MSE loss instead of a general Lipschitz loss. The proof technique however relies on covering numbers rather than Rademacher complexity, which will be key to obtaining a generalization bound for the deep case.

Notice how this bound does not depend on the width $w$, because the $F_1$-norm (and the $F_1$-ball) themselves do not depend on the width. This matches with empirical evidence that shows that increasing the width does not hurt generalization (Belkin et al., 2019; Geiger et al., 2019; 2020).

To use Theorem 1 effectively we need to be able to guarantee that the learned function will have a small enough $F_1$-norm. The $F_1$-norm is hard to compute exactly, but it is bounded by the parameter norm: if $f(x) = W\sigma(Vx + b)$, then $\|f\|_{F_1} \leq \frac{1}{2} \left( \|W\|_F^2 + \|V\|_F^2 + \|b\|^2 \right)$, and this bound is tight if the width $w$ is large enough and the parameters are chosen optimally. Adding weight decay/$L_2$-regularization to the cost then leads to bias towards learning with small $F_1$ norm.

## 3.2 DEEP NETWORKS

Since an AccNet is simply the composition of multiple shallow nets, the functions represented by an AccNet is included in the set of composition of $F_1$ balls. More precisely, if $\|W_\ell\|^2 + \|V_\ell\|^2 + \|b_\ell\|^2 \leq 2R_\ell$ then $f_{L:1}$ belongs to the set $\{g_L \circ \cdots \circ g_1 : g_\ell \in B_{F_1}(0, R_\ell)\}$ for some $R_\ell$.

As noticed in (Bartlett et al., 2017), the covering number is well-behaved under composition, therefore the complexity of AccNets can be bounded in terms of the individual shallow nets it is made of:

**Theorem 2.** *Consider an accordion net of depth $L$ and widths $d_L, \ldots, d_0$, with corresponding set of functions $\mathcal{F} = \{f_{L:1} : \|f_\ell\|_{F_1} \leq R_\ell, Lip(f_\ell) \leq \rho_\ell, \|f_{L:1}\|_\infty \leq B\}$ with input space $\Omega = B(0, r)$. Then with probability $1 - p$ over the sampling of the training set $X$ from the distribution $\pi$, we have for all $f \in \mathcal{F}$*

$$\sqrt{\mathcal{L}(f)} - \sqrt{\tilde{\mathcal{L}}_N(f)} \leq c_0 \sqrt{B\rho_{L:1} r \sum_{\ell'=1}^{L} \frac{R_{\ell'}}{\rho_{\ell'}} \sqrt{d_{\ell'} + d_{\ell'-1}} \frac{1}{\sqrt{N}} (1 + o(1))} + c_1 B \sqrt{\frac{-\log p/2}{N}},$$

*for $c_0 \approx 14.6$ and $c_1 \approx 7.4$. Thus if $\tilde{\mathcal{L}}_N(f) = O(B\rho_{L:1} r \sum_{\ell'=1}^{L} \frac{R_{\ell'}}{\rho_{\ell'}} \sqrt{d_{\ell'} + d_{\ell'-1}} N^{-\frac{1}{2}})$ then $\mathcal{L}(f) = O(B\rho_{L:1} r \sum_{\ell'=1}^{L} \frac{R_{\ell'}}{\rho_{\ell'}} \sqrt{d_{\ell'} + d_{\ell'-1}} N^{-\frac{1}{2}})$.*

Theorem 2 can be extended to ResNets $(f_L + id) \circ \cdots \circ (f_1 + id)$ by simply replacing the Lipschitz constant $Lip(f_\ell)$ by $Lip(f_\ell + id)$.

The Lipschitz constants $Lip(f_\ell)$ are difficult to compute exactly, so it is easiest to simply bound it by the product of the operator norms $Lip(f_\ell) \leq \|W_\ell\|_{op} \|V_\ell\|_{op}$, but we see in Theorem 4 how this bound can be very loose[1] The fact that our bound depends on the Lipschitz constants rather than the operator norms $\|W_\ell\|_{op}, \|V_\ell\|_{op}$ is thus a significant advantage.

---

[1]A simple example of a function whose Lipschitz constant is much smaller than its $F_1$ norm (and thus its operator norms since $\|W\|_{op} \geq \frac{\|W\|_F}{\text{Rank}W}$) is is the 'zig-zag' function $f(x) = |x - a^{-1}\text{round}(ax)|$ on the interval $[0, 1]$: we have $Lip(f) = 1$ but $\|f\|_{F_1} = \Omega(a)$ since one neuron is needed for each up or down, each contributing a constant to the parameter norm.

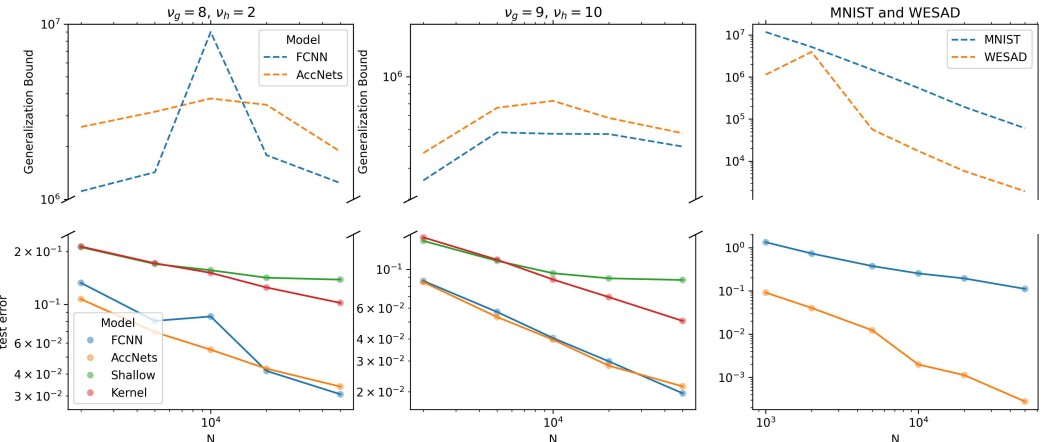

Figure 1: Visualization of scaling laws. We observe that deep networks (either AccNets or DNNs) achieve better scaling laws than kernel methods or shallow networks on certain compositional tasks, in agreement with our theory. We also see that our new generalization bounds approximately recover the right scaling laws (even though they are orders of magnitude too large overall). We consider a compositional true function $f^* = h \circ g$ where $g$ maps from dimension 15 to 3 while h maps from 3 to 20, and we denote $\nu_g, \nu_h$ for the number of times $g, h$ are differentiable. In the first plot $\nu_g = 8, \nu_h = 2$ so that $g$ is easy to learn while $h$ is hard, whereas in the second plot $\nu_g = 9, \nu_h = 10$, so both $g$ and $h$ are relatively easier. The third plot presents the test error and generalization bounds for MNIST and WESAD (Schmidt et al., 2018).

This bound can be applied to a FCNNs with weight matrices $M_1, \ldots, M_{L+1}$, by replacing the middle $M_\ell$ with SVD decomposition $USV^T$ in the middle by two matrices $W_{\ell-1} = \sqrt{S}V^T$ and $V_\ell = U\sqrt{S}$, with dimensions $d_\ell = \mathrm{Rank}M_{\ell+1}$. The Frobenius norm of the new matrices equals the nuclear norm of the original one $\|W_{\ell-1}\|_F^2 = \|V_\ell\|_F^2 = \|M_\ell\|_*$. Recent results have shown that weight-decay leads to a low-rank bias on the weight matrices of the network (Jacot, 2023a;b; Galanti et al., 2022).

This allows us compare our complexity measure to the one in (Bartlett et al., 2017), whose complexity measure $\prod_\ell \|W_\ell\|_{op} \left( \sum_\ell \frac{\|W_\ell\|_{2,1}^{2/3}}{\|W_\ell\|_{op}^{2/3}} \right)^{3/2}$ is closest to ours. We obtain three improvements: replacing the operator norm by the Lipschitz constant of $f_\ell$, replacing the $(2, 1)$-norm ($\|A\|_{2,1} = \|(\|A_{:,1}\|_2, \ldots, \|A_{:,d}\|_2)\|_1$) by the nuclear norm, and removing the $2/3$ and $3/2$ exponents. These three changes play a significant role in Theorem 4, as we discuss later. Note however that since our bound relies on the rank of the weight matrices being bounded, we cannot say that it is a strict improvement over (Bartlett et al., 2017).

We compute in Figure 1 the upper bound of Theorem 2 for both AccNets and DNNs, and even though we observe a very large gap (roughly of order $10^7$), we do observe that it captures rate/scaling of the test error (the log-log slope) well. So this generalization bound could be well adapted to predicting rates, which is what we will do in the next section.

*Remark.* Note that if one wants to compute this upper bound in practical setting, it is important to train with $L_2$ regularization until the parameter norm also converges (this often happens after the train and test loss have converged). The intuition is that at initialization, the weights are initialized randomly, and they contribute a lot to the parameter norm, and thus lead to a larger generalization bound. During training with weight decay, these random initial weights slowly vanish, thus leading to a smaller parameter norm and better generalization bound. It might be possible to improve our generalization bounds to take into account the randomness at initialization to obtain better bounds throughout training, but we leave this to future work.

## 4 BREAKING THE CURSE OF DIMENSIONALITY WITH COMPOSITIONALITY

In this section we study a large family of functions spaces, obtained by taking compositions of Sobolev balls. We focus on this family of tasks because they are well adapted to the complexity measure we have identified, and because kernel methods and even shallow networks do suffer from the curse of dimensionality on such tasks, whereas deep networks avoid it (e.g. Figure 1).

More precisely, we will show that these sets of functions can be approximated by a AccNets with bounded (or in some cases slowly growing) complexity measure

$$R(\theta) = \prod_{\ell=1}^{L} Lip(f_\ell) \sum_{\ell=1}^{L} \frac{\|f_\ell\|_{F_1}}{Lip(f_\ell)} \sqrt{d_\ell + d_{\ell-1}}.$$

This will then allow us show that AccNets can (assuming global convergence) avoid the curse of dimensionality, even in settings that should suffer from the curse of dimensionality, when the input dimension is large and the function is not very smooth (only a few times differentiable).

Since the regularization term $R(\theta)$ is difficult to optimize directly because of the Lipschitz constants, we also consider the upper bound $\tilde{R}(\theta)$ obtained by replacing each $Lip(f_\ell)$ with $\|V_\ell\|_{op}\|W_\ell\|_{op}$.

### 4.1 COMPOSITION OF SOBOLEV BALLS

The family of Sobolev norms capture the regularity of a function, by measuring the size of its derivatives. Consider a function $f : \mathbb{R}^{d_{in}} \to \mathbb{R}$ with derivatives $\partial_x^\alpha f$ for some $d_{in}$-multi-index $\alpha$, the $W^{\nu,p}(\pi)$-Sobolev norm with integer $\nu$ and $p \geq 1$ is defined as

$$\|f\|_{W^{\nu,p}(\pi)}^p = \sum_{|\alpha| \leq \nu} \|\partial_x^\alpha f\|_{L_p(\pi)}^p.$$

Note that the derivative $\partial_x^\alpha f$ only needs to be defined in the 'weak' sense, which means that even non-differentiable functions such as the ReLU functions can actually have finite Sobolev norm.

The Sobolev balls $B_{W^{\nu,p}(\pi)}(0, R) = \{f : \|f\|_{W^{\nu,p}(\pi)} \leq R\}$ are a family of function spaces with a range of regularity (the larger $\nu$, the more regular). This regularity makes these spaces of functions learnable purely from the fact that they enforce the function $f$ to vary slowly as the input changes. Indeed we can prove the following generalization bound:

**Proposition 3.** *Given a distribution $\pi$ with support in $B(0,b)$, we have that with probability $1 - p$ for all functions $f \in \mathcal{F} = \{f : \|f\|_{W^{\nu,2}} \leq R, \|f\|_\infty \leq B\}$*

$$\sqrt{\mathcal{L}(f)} - \sqrt{\tilde{\mathcal{L}}_N(f)} = c_0 \left( \frac{B^2 R^r}{N} \right)^{\frac{1}{2+r}} + c_1 B \sqrt{\frac{-\log p/2}{N}},$$

*where $r = \frac{d_{in}}{\nu}$. Therefore if $\tilde{\mathcal{L}}_N(f) = O((\frac{B^2 R^r}{N})^{\frac{2}{2+r}})$, then $\mathcal{L}(f) = O((\frac{B^2 R^r}{N})^{\frac{2}{2+r}})$.*

But this result also illustrates the **curse of dimensionality:** the differentiability $\nu$ needs to scale with the input dimension $d_{in}$ to obtain a reasonable rate. If instead $\nu$ is constant and $d_{in}$ grows, then the number of datapoints $N$ needed to guarantee a test error of at most $\epsilon^2$ scales exponentially in $d_{in}$, i.e. $N \sim \epsilon^{-(2+r)} = \epsilon^{-(2+\frac{d_{in}}{\nu})}$. One way to interpret this issue is that regularity becomes less and less useful the larger the dimension: knowing that similar inputs have similar outputs is useless in high dimension where the closest training point $x_i$ to a test point $x$ is typically very far away.

#### 4.1.1 BREAKING THE CURSE OF DIMENSIONALITY WITH COMPOSITIONALITY

To break the curse of dimensionality, we need to assume some additional structure on the data or task which introduces an 'intrinsic dimension' that can be much lower than the input dimension $d_{in}$:

**Manifold hypothesis**: If the input distribution lies on a $d_{surf}$-dimensional manifold, the error rates typically depends on $d_{surf}$ instead of $d_{in}$ (Schmidt-Hieber, 2019; Chen et al., 2022).

**Known Symmetries:** If $f^*(g \cdot x) = f^*(x)$ for a group action $\cdot$ w.r.t. a group $G$, then $f^*$ can be written as the composition of a quotient map $g^* : \mathbb{R}^{d_{in}} \to \mathbb{R}^{d_{in}}/G$ which maps pairs of inputs which

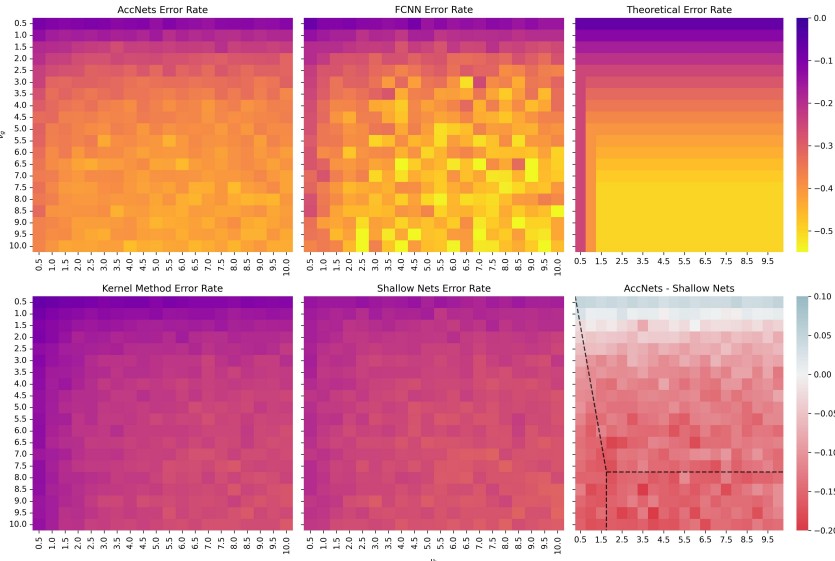

Figure 2: A comparison of empirical and theoretical error rates. The frist two columns show the log decay rate of the test error with respect to the dataset size $N$ based on our empirical simulations for 4 different models. The top right plot depicts the theoretical decay rate of the test error $-\min\{\frac{1}{2}, \frac{2\nu_g}{2\nu_g+d_{in}}, \frac{2\nu_h}{2\nu_h+d_{mid}}\}$. The bottom right plot displays the difference between the rates of AccNets and shallow nets. The lower left region represents the area where $g$ is easier to learn than $h$, the upper right where $h$ is easier to learn than $g$, and the lower right region where both $f$ and $g$ are easy. We see that the biggest gain of AccNets over shallow nets are in the. lower left regions, where learning $h$ is hard.

are equivalent up to symmetries to the same value (pairs $x, y$ s.t. $y = g \cdot x$ for some $g \in G$), and then a second function $h^* : \mathbb{R}^{d_{in}}/G \to \mathbb{R}^{d_{out}}$, then the complexity of the task will depend on the dimension of the quotient space $\mathbb{R}^{d_{in}}/G$ which can be much lower. If the symmetry is known, then one can for example fix $g^*$ and only learn $h^*$ (though other techniques exist, such as designing kernels or features that respect the same symmetries) (Mallat, 2012).

**Symmetry Learning:** However if the symmetry is not known then both $g^*$ and $h^*$ have to be learned, and this is where we require feature learning and/or compositionality. Shallow networks are able to learn translation symmetries, since they can learn so-called low-index functions which satisfy $f^*(x) = f^*(Px)$ for some projection $P$ (with a statistical complexity that depends on the dimension of the space one projects into, not the full dimension (Bach, 2017; Abbe et al., 2021)). Low-index functions correspond exactly to the set of functions that are invariant under translation along the kernel $\ker P$. To learn general symmetries, one needs to learn both $h^*$ and the quotient map $g^*$ simultaneously, hence the importance of feature learning.

For $g^*$ to be learnable efficiently, it needs to be regular enough to not suffer from the curse of dimensionality, but many traditional symmetries actually have smooth quotient maps, for example the quotient map $g^*(x) = \|x\|^2$ for rotation invariance. This can be understood as a special case of composition of Sobolev functions, whose generalization gap can be bounded:

**Theorem 4.** *Let* $\mathcal{F} = \mathcal{F}_L \circ \cdots \circ \mathcal{F}_1$ *where* $\mathcal{F}_\ell = \{f_\ell : \mathbb{R}^{d_{\ell-1}} \to \mathbb{R}^{d_\ell} \text{ s.t. } \|f_\ell\|_{W^{\nu_\ell,2}} \leq R, \|f_\ell\|_\infty \leq B, Lip(f_\ell) \leq \rho_\ell\}$, *and let* $r_{\max} = \max_\ell r_\ell$ *for* $r_\ell = \frac{d_{\ell-1}}{\nu_\ell}$, *then with probability* $1 - p$ *we have for all* $f \in \mathcal{F}$

$$\sqrt{\mathcal{L}(f)} - \sqrt{\tilde{\mathcal{L}}_N(f)} \leq c_0 \left(\frac{B^2 R^{r_{max}}}{N}\right)^{\frac{1}{2+r_{max}}} + c_1 B \sqrt{\frac{-\log p}{N}},$$

*where* $c_0$ *depends only on* $r_\ell, \rho_\ell$. *Thus if* $\tilde{\mathcal{L}}_N(f) \leq O(N^{-\frac{2}{2+r_{max}}})$ *then* $\mathcal{L}(f) \leq O(N^{-\frac{2}{2+r_{max}}})$.

We see that only the largest ratio $r_{\max}$ matters when it comes to the rate of learning. Coming back to the symmetry learning example, we see that the hardness of learning a function of the type $f^* = h \circ g$ with inner dimension $d_{mid}$ and regularities $\nu_g$ and $\nu_h$ leading to ratios $r_g = \frac{d_{in}}{\nu_g}$ and $r_h = \frac{d_{mid}}{\nu_h}$, the test error will of order (assuming the train error is small enough)

$$\mathcal{L} = O\left(N^{-\min\{\frac{2}{2+r_g}, \frac{2}{2+r_h}\}}\right) = O\left(N^{-\min\{\frac{2\nu_g}{2\nu_g+d_{in}}, \frac{2\nu_h}{2\nu_h+d_{mid}}\}}\right).$$

This suggests the existence of two regimes depending on whether $g$ or $h$ is harder to learn, which is determined by which ratio $r_g$ or $r_h$ is larger.

In contrast, without taking advantage of the compositional structure, we expect $f^*$ to be only $\min\{\nu_g, \nu_h\}$ times differentiable, so trying to learn it as a single Sobolev function would lead to an error rate of $N^{-\frac{2\min\{\nu_g, \nu_h\}}{2\min\{\nu_g, \nu_h\}+d_{in}}} = N^{-\min\{\frac{2\nu_g}{2\nu_g+d_{in}}, \frac{2\nu_h}{2\nu_h+d_{in}}\}}$ which is no better than the compositional rate assuming $d_{mid} \leq d_{in}$, and can in some cases be arbitrarily worse.

Furthermore, since multiple compositions are possible, one can imagine a hierarchy of symmetries that slowly reduce the dimensionality with less and less regular quotient maps. For example one could imagine a composition $f_L \circ \cdots \circ f_1$ with dimensions $d_\ell = d_0 2^{-\ell}$ and regularities $\nu_\ell = d_0 2^{-\ell}$ so that the ratios remain constant $r_\ell = \frac{d_0 2^{-\ell}}{d_0 2^{-\ell+1}} = \frac{1}{2}$, leading to an almost parametric rate of $N^{-\frac{1}{2}} \log N$, improving significantly over to the non-compositional rate of $N^{-2^{-L}}$.

*Remark.* A naive argument suggests that the rate of $N^{-\frac{2}{2+r_{\max}}}$ should be optimal: assume that the maximum $r_\ell$ is attained at a layer $\ell$, then one can consider the subset of functions such that the image $f_{\ell-1:1}(B(0,r))$ contains a ball $B(z, r') \subset \mathbb{R}^{d_\ell-1}$ and that the function $f_{L:\ell+1}$ is $\beta$-non-contracting $\|f_{L:\ell+1}(x) - f_{L:\ell+1}(y)\| \geq \beta \|x - y\|$, then learning $f_{L:1}$ should be as hard as learning $f_\ell$ over the ball $B(z, r')$, thus forcing a rate of at least $N^{-\frac{2}{2+r_{\max}}}$. An analysis of minimax rates in a similar setting has been done in (Giordano et al., 2022).

## 4.2 Breaking the Curse of Dimensionality with AccNets

Now that we know that composition of Sobolev functions can be easily learnable, even in settings where the curse of dimensionality should make it hard to learn them, we need to find a model that can achieve those rates. Though many models are possible [2], we focus on DNNs, in particular AccNets. Assuming convergence to a global minimum of the loss of sufficiently wide AccNets with two types of regularization, one can guarantee close to optimal rates:

**Theorem 5.** *Given a true function $f^*_{L^*:1} = f^*_{L^*} \circ \cdots \circ f^*_1$ going through the dimensions $d^*_0, \ldots, d^*_{L^*}$, along with a continuous input distribution $\pi_0$ supported in $B(0, b_0)$, such that the distributions $\pi_\ell$ of $f^*_\ell(x)$ (for $x \sim \pi_0$) are continuous too and supported inside $B(0, b_\ell) \subset \mathbb{R}^{d^*_\ell}$. Further assume that there are differentiabilities $\nu_\ell$ and radii $R_\ell$ such that $\|f^*_\ell\|_{W^{\nu_\ell, 2}(B(0, b_\ell))} \leq R_\ell$, and $\rho_\ell$ such that $Lip(f^*_\ell) \leq \rho_\ell$. For an infinite width AccNet with $L \geq L^*$ and dimensions $d_\ell \geq d^*_1, \ldots, d^*_{L^*-1}$, we have for the ratios $\tilde{r}_\ell = \max\{\frac{d^*_{\ell-1}+3}{\nu_\ell}, 2\}$:*

*(1) At a global min $\hat{f}_{L:1}$ of $\tilde{\mathcal{L}}_N(f_{L:1}) + \lambda R(\theta)$, we have $\mathcal{L}(\hat{f}_{L:1}) = O(N^{-\frac{2}{2+\tilde{r}_{\max}}})$ for $\tilde{r}_{\max} = \max\{\tilde{r}_1, \ldots, \tilde{r}_L\}$.*

*(2) At a global min $\hat{f}_{L:1}$ of $\tilde{\mathcal{L}}_N(f_{L:1}) + \lambda \tilde{R}(\theta)$, we have $\mathcal{L}(\hat{f}_{L:1}) = O(N^{-\frac{2}{2+\tilde{r}_{sum}}})$ for $\tilde{r}_{sum} = 2 + \sum_\ell (\tilde{r}_\ell - 2)$.*

In the proof, we build a network $\hat{f}_{L:1}$ that approximates $f^*_{L^*:1}$ in the following manner: we adapt approximation results from (Bach, 2017) to approximate each Sobolev function $f^*_\ell$ by a shallow network $\hat{f}_{\ell'}$ and if $L > L^*$ we add $L - L^*$ identity layers where the dimensionality $d^*_\ell$ is minimal (the parameter norm required to represent the identity on a $d$-dimensional representation is proportional to $d$, so it is optimal to add identity layers at the smallest dimension).

---

[2] One could argue that it would be more natural to consider compositions of kernel method models, for example a composition of random feature models. But this would lead to a very similar model: this would be equivalent to a AccNet where only the $W_\ell$ weights are learned, while the $V_\ell, b_\ell$ weights remain constant. Another family of models that should have similar properties is Deep Gaussian Processes (Damianou & Lawrence, 2013).

$F_1$**-norm/Barron norm:** This shows the power of replacing other parameter norm measures (such as the $(2, 1)$-norm from Bartlett et al. (2017)) with the $F_1$-norm, by allowing us to repurpose the already existing approximation results for shallow networks. While we do not prove that a similar approximations cannot be achieved with small $(2, 1)$-norm too, it seems unlikely, since generic Sobolev functions require a high variety of active neurons (a natural way to approximate them is to use a random feature approach (Rahimi & Recht, 2008)) which seems in contradiction with the $(2, 1)$-norm which imposes a sparsity on the neurons.

**Norm based vs Sparsity based:** In (Schmidt-Hieber, 2020; Poggio & Fraser, 2024) a similar class of functions is considered and optimal rates $N^{-\frac{2}{2+r_{max}}}$ with $r_{max} = \max\{\frac{d_0^*}{\nu_1}, \dots, \frac{d_{L^*-1}^*}{\nu_{L^*}}\}$ instead of $\tilde{r}_{max} = \max\{\frac{d_0^*+3}{\nu_1}, \dots, \frac{d_{L^*-1}^*+3}{\nu_{L^*}}, 2\}$ are proven. The presence off the $+3$ comes from the shallow network approximation results and could potentially be removed. The bigger gap happens for easy tasks, where the 'fast' rates can be achieved $N^{-1}$ when $r_{max} \approx 0$, whereas our result can never predict rates faster than $N^{-\frac{1}{2}}$ because $\tilde{r}_{max} \geq 2$. This might be a fundamental gap, because norm-based complexity measures are known to be unable to yield fast rates (Srebro et al., 2010), so a control on the dimensionality of the model might be required to obtain these fast rates. Real world tasks are probably rarely easy enough to achieve rates faster than $N^{-\frac{1}{2}}$, e.g. LLMs scaling laws have been observed to be roughly $N^{-0.095}$ in (Kaplan et al., 2020).

**Lipschitz constant vs operator norm:** By evaluating the complexities $R, \tilde{R}$ on this approximation, we obtain two bounds illustrating the tradeoff between these two complexities. The first complexity measure based on Lipschitz constant $Lip(f_\ell)$ leads to almost optimal rates, whereas the second complexity measure can lead to arbitrarily suboptimal rates when two or more ratios $\frac{d_{\ell-1}^*+3}{\nu_\ell}$ are larger than 2. This is due to the fact to approximate a general Sobolev functions with a large ratio $\frac{d_{\ell-1}^*+3}{\nu_\ell}$ with a shallow network, the parameter norm $\|W_\ell\|_F^2 + \|V_\ell\|_F^2$ needs to grow to infinity as $\epsilon \searrow 0$, leading to exploding operator norms, even though the Lipschitz constant remains bounded.

Obviously, the Lipschitz constants $Lip(f_\ell)$ are difficult to optimize over. For finite width networks it is in theory possible to take the max over all linear regions, but the complexity might be unreasonable. It might be more reasonable to leverage an implicit bias instead, such as a large learning rate, because a large Lipschitz constant implies that the network is sensible to small changes in its parameters, so GD with a large learning rate should only converge to minima with a small Lipschitz constant (such a bias is described in (Jacot, 2023b)). For this reason, we apply standard weight-decay to minimize the $F_1$-norms, and rely on the implicit bias to control the Lipschitz constants. Going further, techniques from (Wei & Ma, 2019) could be used to replace the Lipschitz constant in the bound with the maximum Jacobian over the dataset, which could then be optimized directly.

**Impact of Depth:** As can be seen in the proof of Theorem 5, when the depth $L$ is strictly larger than the true depth $L^*$, one needs to add identity layers, leading to a so-called Bottleneck structure, which was proven emerge as a result of weight decay in (Jacot, 2023a;b; Wen & Jacot, 2024). These identity layers add a term that scales linearly in the additional depth $\frac{(L-L^*)d_{min}^*}{\sqrt{N}}$ to the first regularization, (and multiplicative factor of order $L$ to the second), see the proof for more details. The removal of the $2/3$ and $3/2$ exponents[3] in our bounds in comparison to previous bounds Bartlett et al. (2017) allow us to get such a $O(L)$ bound instead of $O(L^{\frac{3}{2}})$. Interestingly, the first bound proposed in Golowich et al. (2020) to obtain a better depth dependence depends on the product of the Frobenius norms, which would grow exponentially as $\min_\ell\{d_0^*, \dots, d_L^*\}^{(L-L^*)}$ if the minimum dimension is 2 or more. Finally, note that by switching to a ResNet, there is no need for those identity layers, and the generalization becomes constant in depth.

**Limitations:** There are a number of limitations to this result. First we assume that one is able to recover the global minimizer of the regularized loss, which should be hard in general[4] (we already know from (Bach, 2017) that this is NP-hard for shallow networks and a simple $F_1$-regularization). Note that it is sufficient to recover a network $f_{L:1}$ whose regularized loss is within a constant of the

---

[3]We achieve this by applying chaining accross the layers instead of applying to each layer separately.

[4]Note that the unregularized loss can be optimized polynomially, e.g. in the NTK regime (Jacot et al., 2018; Allen-Zhu et al., 2019; Du et al., 2019), but this is an easier task than finding the global minimum of the regularized loss where one needs to both fit the data, and do it with an minimal regularization term.

global minimum, which might be easier to guarantee, but should still be hard in general. The typical method of training with GD on the regularized loss is a greedy approach, which might fail in general but could recover almost optimal parameters under the right conditions (some results suggest that training relies on first order correlations to guide the network in the right direction (Abbe et al., 2021; 2022; Petrini et al., 2023)).

Another limitation is that our proof requires an infinite width, because we were not able to prove that a function with bounded $F_1$-norm and Lipschitz constant can be approximated by a sufficiently wide shallow networks with the same (or close) $F_1$-norm and Lipschitz constant (we know from (Bach, 2017) that it is possible without preserving the Lipschitzness). We are quite hopeful that this condition might be removed in future work.

### 4.2.1 EXPERIMENTS

We train our models on a synthetic dataset obtained by composing two Gaussian processes $Y = h(g(X))$ with Matérn kernels $K_g, K_h$ chosen so that $g$ and $h$ have the right differentiability. In Figure 2, we compare the empirical rates (by doing a linear fit on a log-log plot of test error as a function of $N$) and rates $\min\{\frac{1}{2}, \frac{2\nu_g}{2\nu_g+d_{in}}, \frac{2\nu_h}{2\nu_h+d_{mid}}\}$ which seem to yield the best match. Note that this best fit for the rates is a mix of the theoretical lower bound $\min\{\frac{2\nu_g}{2\nu_g+d_{in}}, \frac{2\nu_h}{2\nu_h+d_{mid}}\}$ and the upper bound $\min\{\frac{1}{2}, \frac{2\nu_g}{2\nu_g+d_{in}+3}, \frac{2\nu_h}{2\nu_h+d_{mid}+3}\}$, suggesting that the appearance of the $+3$ in the denominator might be an artifact of the proofs, whereas the $\frac{1}{2}$ might actually be optimal. But the amount of noise in our approximation of the rates does not allow us to give a definite answer for what the rates should be.

Notice also that all experiments were done with a simple $L_2$-regularization, which suggests the Lipshitz constants $Lip(f_\ell)$ do not need to be optimized directly. The use of large learning rates and the edge of stability phenomenon (Cohen et al., 2021) could explain this implicit control of the Lipschitzness, because a large Lipschitz constant can lead to exploding gradients, and training with large learning rates could implicit avoid such parameters. Stability under large learning rate has been shown to control the regularity of the representations in previous works (Jacot, 2023a;b).

### 4.2.2 COMPUTATIONAL GRAPHS

Theorem 5 can be adapted to show that AccNets can also learn any 'computational graph' as in (Schmidt-Hieber, 2020; Poggio & Fraser, 2024), where a variables $v_1, \ldots, v_K$ each with their own dimension $d_k$. Each variable has a set of parents $P(k) \subset \{1, \ldots, K\}$ on whom it depends $v_k = f_k((v_m : m \in P(k)))$ for a $\nu_k$-Sobolev or $F_1$ function $g_k$. We assume that the variables can be ordered in such a way that all parents of a variable belong have smaller indices, or equivalently that the directed graph with $K$ vertices, with edges connecting parents to their child is acyclic. The input variable is $1$ and output one is $K$.

It is then possible to organize the variables into 'layers' of variables that only have parents in the previous layers (possibly adding identities to keep certain variables from previous layers until they are needed). We say that the depth $L^*$ of the computational graph is the minimal number of layers required to find such a representation, and it matches the length of the longest directed path in the graph. Thus the complete function can be written as the composition of $L^*$ functions $f^*_{L:1}$, where each $f^*_\ell$ is the product of a few $g_k$s, e.g. $f^*_\ell(v_1, v_2, v_3) = (g_1(v_1, v_2), g_2(v_3), g_3(v_1, v_3))$. The $F_1$-norm has the property that $F_1$-norm of the product of multiple functions is upper bounded by the sum of the $F_1$ norms, e.g. $\|f^*_\ell\|_{F_1} = \|g_1\|_{F_1} + \|g_2\|_{F_1} + \|g_3\|_{F_1}$, as well as the Lipshchitz constant $Lip(f^*_\ell) = Lip(g_1) + Lip(g_2) + Lip(g_3)$. One can then easily see that, assuming bounded Lipschitz constants, we can guarantee a rate of learning of $N^{-\frac{2}{2+\tilde{r}_{\max}}}$ for $\tilde{r}_{max} = \max_{k=1,\ldots,K} \frac{3+\sum_{m \in P(k)} d_k}{\nu_k}$.

## 5 CONCLUSION

We have given a generalization bound for Accordion Networks and as an extension Fully-Connected networks. It depends on the $F_1$-norms and Lipschitz constants of its shallow subnetworks. This allows us to prove under certain assumptions that AccNets can learn general compositions of Sobolev functions efficiently, making them able to break the curse of dimensionality in certain settings, such as in the presence of unknown symmetries.

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

# A  EXPERIMENTAL SETUP

In this section, we review our numerical experiments and their setup both on synthetic and real-world datasets in order to address theoretical results more clearly and intuitively. The code used for experiments is publicly available here.

## A.1  DATASET

### A.1.1  EMPIRICAL DATASET

By Corollary A.6 from (Tuo & Wu, 2015), the Reproducing kernel Hilbert space (RKHS) generated by Matérn kernel is norm equivalent to the Sobolev space $H^{\nu+d/2}(\Omega)$, so we utilized Matérn kernels to generate the synthetic dataset which retains Sobolev properties.

The Matérn kernel is considered a generalization of the radial basis function (RBF) kernel. It controls the differentiability, or smoothness, of the kernel through the parameter $\nu$. As $\nu \to \infty$, the Matérn kernel converges to the RBF kernel, and as $\nu \to 0$, it converges to the Laplacian kernel, a 0-differentiable kernel. In this study, we utilized the Matérn kernel to generate Gaussian Process (GP) samples based on the composition of two Matérn kernels, $K_g$ and $K_h$, with varying differentiability in the range [0.5,10]×[0.5,10]. The input dimension ($d_{in}$) of the kernel, the bottleneck mid-dimension ($d_{mid}$), and the output dimension ($d_{out}$) are 15, 3, and 20, respectively.

This outlines the general procedure of our sampling method for synthetic data:

1. Sample the training dataset $X \in \mathbb{R}^{D \times d_{in}}$

2. From X, compute the $D \times D$ kernel $K_g$ with given $\nu_g$

3. From $K_g$, sample $Z \in \mathbb{R}^{D \times d_{mid}}$ with columns sampled from the Gaussian $\mathcal{N}(0, K_g)$.

4. From Z, compute $K_g$ with given $\nu_h$

5. From $K_h$, sample the test dataset $Y \in \mathbb{R}^{D \times d_{out}}$ with columns sampled from the Gaussian $\mathcal{N}(0, K_h)$.

We used 128 GB of RAM to generate our synthetic dataset. Due to the Matérn Kernel's time complexity of $\mathcal{O}(n^3)$ and the space complexity of $\mathcal{O}(n^2)$, the maximum possible dataset size for 128 GB of memory is approximately 52,500. Among the 52,500 dataset points, 50,000 were allocated for training, and 2,500 were used for the test dataset.

### A.1.2  REAL-WORLD DATASET: MNIST AND WESAD

In our study, we utilized both MNIST (Modified National Institute of Standards and Technology) and WESAD (Wearable Stress and Affect Detection) to train our AccNets for classification tasks. As a standard benchmark for image classification tasks, MNIST consists of 60,000 grayscale images of 10 different handwritten digits. On the other hand, the WESAD dataset provides multimodal physiological and motion data collected from 15 subjects using devices worn on the wrist and chest. For the purpose of our experiment, we specifically employed the Empatica E4 wrist device to distinguish between non-stress (baseline) and stress conditions.

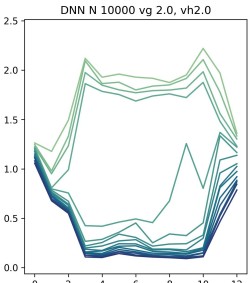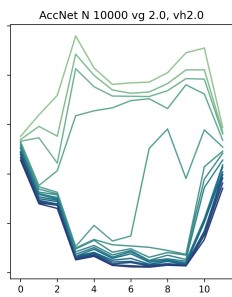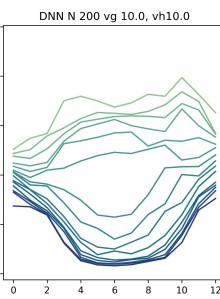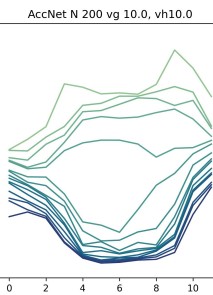

Figure 3: A comparison: singular values of the weight matrices for DNN and AccNets models. The first two plots represent cases where $N = 10000$ while the right two plots correspond to $N = 200$. The number of outliers at the top of each plot signifies the rank of each network. The plots with $N = 10000$ datasets demonstrate a clearer capture of the true rank compared to those with $N = 200$ indicating that a higher dataset count provides more accurate rank determination.

After preprocessing, the dataset comprised a total of 136,482 instances. We implemented a train-test split ratio of approximately 75:25, resulting in 100,000 dataset for the training set and the rest 36,482 dataset for the test set. The overall hyperparameters and architecture of the AccNets model applied to the WESAD dataset were largely consistent with those used for our synthetic data. The primary differences were the use of 100 epochs for each iteration of $N_i$ from $N$ in the model setup and a learning rate set to 1e-5.

## A.2 MODEL SETUPS

For analyzing the scaling law of test error for our synthetic dataset, we trained models with gradually increasing $N_i$ data points from our training data, where $N = [100, 200, 500, 1000, 2000, 5000, 10000, 20000, 50000]$. The models utilized for our analysis are the kernel method, shallow networks, fully connected deep neural networks (FCNN), and Acc-Nets. For FCNN and AccNets, we set the network depth to 12 layers, with the layer widths as $[d_{in}, 500, 500, ..., 500, d_{out}]$ for DNNs, and $[d_{in}, 900, 100, 900, ..., 100, 900, d_{out}]$ for AccNets.

In order to have a comparable number of neurons between the shallow networks and other deeper networks, the width for the shallow networks was set to 50,000, resulting in dimensions of $[d_{in}, 50000, d_{out}]$.

We utilized ReLU as the activation function and $L^2$-norm as the cost function, with the Adam optimizer. The total number of batch was set to 5, and the training process was conducted over 3600 epochs, divided into three phases. The detailed optimizer parameters are as follows:

1. For the first 1200 epochs: learning rate $(lr) = 1.5 * 0.001$, weight decay $= 0$

2. For the second 1200 epochs: $lr = 0.4 * 0.001$, weight decay $= 0.002$

3. For the final 1200 epochs: $lr = 0.1 * 0.001$, weight decay $= 0.005$

We conducted experiments utilizing 12 NVIDIA V100 GPUs (each with 32GB of memory) over approximately 7 days to train all the synthetic datasets. In contrast, training the WESAD and MNIST dataset required only one hour on a single V100 GPU.

## B COVERING NUMBERS

We will use a covering number argument to prove generalization bounds. Before we start, we must define a few notions.

The $\epsilon$-covering number $\mathcal{N}_2(\mathcal{F}; \epsilon)$ of set $\mathcal{F}$ of functions $f : \Omega \to \mathbb{R}^d$, is the smallest integer $n$ such that for any distribution $\pi$ supported on $\Omega$ there is a covering $\tilde{\mathcal{F}}$, i.e. a finite set with no more than $n$

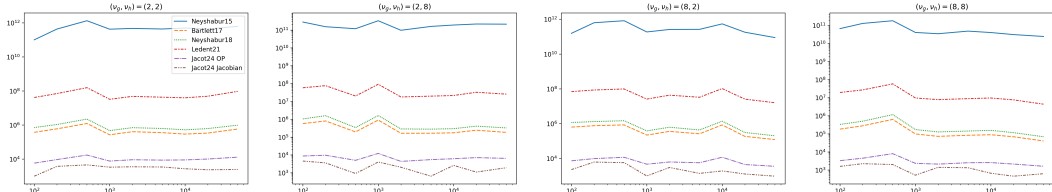

Figure 4: Comparison of our bounds to a number of previous bounds on the composition of two functions dataset across different differentiabilities of $g$ and $h$. For simplicity, we drop the constant prefactors in each bound, since they are mostly an artifact of the proofs. We see that our bound (Jacot24 Jacobian) obtains strictly better than previous ones, even when using upper bounding the Lipschitz constants by the operator norms (Jacot24 OP). Since we cannot compute the Lipschitz constant, we approximate them by a max over 100 random points (taking more points does not significantly change the final bound). The lines appear flat because most of the variation is between different bounds rather than as $N$ increases, but by zooming in, we can see a downward trend, albeit a slow and noisy one. The case $(2, 2)$ lies in the regime where both $g$ and $h$ are hard, in which case our theory predicts that the operator norm bound should have a worse scaling exponent than the Lipschitz based bound. The experiments appear to match our prediction, because the Lipschitz-constant based bound seems to be the only one that is decreasing in $N$, while the operator norm bound, as well as all previous bounds, are increasing (the fact that uniform generalization bounds can be increasing in $N$ has been observed in previous work (Nagarajan & Kolter, 2019)). Though these trends are interesting, there is clearly a lot of noise in the resulting curves, so it is difficult to confidently conclude anything from these experiments.

elements, such that for all $f \in \mathcal{F}$ there is a cover $\tilde{f} \in \tilde{\mathcal{F}}$ with

$$\left\| f - \tilde{f} \right\|_\pi = \sqrt{\mathbb{E}_{x \sim \pi} \left\| f(x) - \tilde{f}(x) \right\|^2} \leq \epsilon.$$

We will also sometimes say that $\tilde{f}$ covers $f$ in such case.

The covering number allows us to bound the difference between the errors of two datasets uniformly over the whole functions set $\mathcal{F}$ with high-probability:

**Theorem 6.** *Let $\mathcal{F}$ be a space of functions such that $\|g\|_\infty \leq B$ for all $g \in \mathcal{F}$. Given a true function $f^*$ such that $\|f\|_\infty \leq B$, we have that with probability at least $1 - p$ over the sampling of two datasets $S, S'$ from a distribution $\pi$, one has for all $g \in \mathcal{F}$:*

$$\|g - f^*\|_{S'} - \|g - f^*\|_S \leq 2 \min_\epsilon \epsilon + B\sqrt{2 \frac{\log \mathcal{N}(\mathcal{F}, \epsilon) - \log p}{N}}.$$

*Proof.* We start with a Chernoff bound:

$$\mathbb{P}_{S,S'} \left[ \sup_{g \in \mathcal{F}} \|g - f^*\|_{S'} - \|g - f^*\|_S \geq c \right] \leq \mathbb{E}_{S,S'} \left[ \exp \left( t \sup_{g \in \mathcal{F}} \|g - f^*\|_{S'} - \|g - f^*\|_S \right) \right] e^{-tc}$$

Defining $M(t) = \mathbb{E}_{S,S'} \left[ \exp \left( t \sup_{g \in \mathcal{F}} \|g - f^*\|_{S'} - \|g - f^*\|_S \right) \right]$, this can be reformulated as saying that with probability $1 - p$, we have that for all $g \in \mathcal{F}$

$$\|g - f^*\|_{S'} - \|g - f^*\|_S < \min_t \frac{\log M(t) - \log p}{t}.$$

Let us now bound $M(t)$. For any pair of datasets $S, S'$, we consider the $\epsilon$-covering $\tilde{\mathcal{F}}$ of $\mathcal{F}$ w.r.t. to the measure $\frac{S+S'}{2}$ which is the empirical measure corresponding to the dataset of size $2N$ made up of the union of $S$ and $S'$. Thus for any function $g \in \mathcal{F}$, there is a $\tilde{g} \in \tilde{\mathcal{F}}$ such that $\|g - \tilde{g}\|_{\frac{S+S'}{2}} \leq \epsilon$ and $\left| \tilde{\mathcal{F}} \right| \leq \mathcal{N}(\mathcal{F}, \epsilon)$.

We can now bound

$$M(t) = \mathbb{E}_{S,S'} \left[ \exp\left( t \sup_{g \in \mathcal{F}} \|g - f^*\|_{S'} - \|g - f^*\|_S \right) \right]$$

$$\leq \mathbb{E}_{S,S'} \left[ \exp\left( t \sup_{g \in \mathcal{F}} \|\tilde{g} - g\|_S + \|\tilde{g} - g\|_{S'} + \|\tilde{g} - f^*\|_{S'} - \|\tilde{g} - f^*\|_S \right) \right]$$

$$\leq e^{2t\epsilon} \mathbb{E}_{S,S'} \left[ \exp\left( t \sup_{g \in \mathcal{F}} \|\tilde{g} - f^*\|_{S'} - \|\tilde{g} - f^*\|_S \right) \right]$$

$$= e^{2t\epsilon} \mathbb{E}_{S,S'} \left[ \sup_{g \in \mathcal{F}} \exp\left( t \frac{\sum_{i=1}^N \|\tilde{g}(x_i) - f^*(x_i)\|^2 - \|\tilde{g}(x_i') - f^*(x_i')\|^2}{N \left( \|\tilde{g} - f^*\|_{S'} + \|\tilde{g} - f^*\|_S \right)} \right) \right]$$

$$\leq e^{2t\epsilon} \mathbb{E}_{S,S'} \left[ \sup_{g \in \mathcal{F}} \exp\left( t \frac{\sum_{i=1}^N \|\tilde{g}(x_i) - f^*(x_i)\|^2 - \|\tilde{g}(x_i') - f^*(x_i')\|^2}{\sqrt{2} N \|\tilde{g} - f^*\|_{\frac{S+S'}{2}}} \right) \right]$$

where we used the fact that $\|h\|_S + \|h\|_{S'} \leq \sqrt{2\|h\|_S^2 + 2\|h\|_{S'}^2} = 2\|h\|_{\frac{S+S'}{2}} \leq 2\epsilon$ and conversely $\|h\|_S + \|h\|_{S'} \geq \sqrt{\|h\|_S^2 + \|h\|_{S'}^2} = \sqrt{2}\|h\|_{\frac{S+S'}{2}}$. We may now introduce i.i.d. Rademacher random variables $\sigma_i$ equal to 1 and $-1$ with prob. $\frac{1}{2}$:

$$M(t) \leq e^{2t\epsilon} \mathbb{E}_{S,S'} \left[ \mathbb{E}_\sigma \left[ \sup_{\tilde{g} \in \tilde{\mathcal{F}}} \exp\left( t \frac{\sum_{i=1}^N \sigma_i \left( \|\tilde{g}(x_i) - f^*(x_i)\|^2 - \|\tilde{g}(x_i') - f^*(x_i')\|^2 \right)}{\sqrt{2} N \|\tilde{g} - f^*\|_{\frac{S+S'}{2}}} \right) \right] \right]$$

$$\leq e^{2t\epsilon} \mathbb{E}_{S,S'} \left[ \sum_{\tilde{g} \in \tilde{\mathcal{F}}} \prod_{i=1}^N \mathbb{E}_{\sigma_i} \left[ \exp\left( \frac{t\sigma_i \left( \|\tilde{g}(x_i) - f^*(x_i)\|^2 - \|\tilde{g}(x_i') - f^*(x_i')\|^2 \right)}{\sqrt{2} N \|\tilde{g} - f^*\|_{\frac{S+S'}{2}}} \right) \right] \right].$$

To understand why adding the Rademacher random variables $\sigma_i$ does not change the expectation, one can think of these random variables as randomly switching the $i$-th datapoints $x_i$ from $S$ and $x_i'$ from $S'$ with each other (not that these switches leave the denominator unchanged too). Sampling two datasets $S$ and $S'$ followed by random switches is equivalent to sampling without the switches.

We now use Hoeffdings Lemma, which tells us that for any random var $X$ with $\mathbb{E}X = 0$ and $X \in [-b, b]$ a.s. one has $\mathbb{E} \exp tX \leq \exp \frac{t^2 b^2}{2}$ to obtain

$$M(t) \leq e^{2t\epsilon} \mathbb{E}_{S,S'} \left[ \sum_{\tilde{g} \in \tilde{\mathcal{F}}} \exp\left( \frac{t^2 \sum_{i=1}^N \left( \|\tilde{g}(x_i) - f^*(x_i)\|^2 - \|\tilde{g}(x_i') - f^*(x_i')\|^2 \right)^2}{2N^2 \|\tilde{g} - f^*\|_{\frac{S+S'}{2}}^2} \right) \right]$$

$$\leq e^{2t\epsilon} \mathbb{E}_{S,S'} \left[ \sum_{\tilde{g} \in \tilde{\mathcal{F}}} \exp\left( \frac{t^2 B^2 \sum_{i=1}^N \|\tilde{g}(x_i) - f^*(x_i)\|^2 + \|\tilde{g}(x_i') - f^*(x_i')\|^2}{N^2 \|\tilde{g} - f^*\|_{\frac{S+S'}{2}}^2} \right) \right]$$

$$= e^{2t\epsilon} \mathbb{E}_{S,S'} \left[ \sum_{\tilde{g} \in \tilde{\mathcal{F}}} \exp\left( \frac{t^2 B^2 \left( \|\tilde{g} - f^*\|_S^2 + \|\tilde{g} - f^*\|_{S'}^2 \right)}{N \|\tilde{g} - f^*\|_{\frac{S+S'}{2}}^2} \right) \right]$$

$$\leq e^{2t\epsilon} \mathbb{E}_{S,S'} \left[ \sum_{\tilde{g} \in \tilde{\mathcal{F}}} \exp\left( \frac{t^2 2B^2}{N} \right) \right]$$

$$\leq \mathcal{N}(\mathcal{F}, \epsilon) \exp\left( 2t\epsilon + \frac{t^2 2B^2}{N} \right).$$

This bound on $M(t)$ then implies that with probability at least $1 - p$ over the sampling of $S, S'$ for all $g \in \mathcal{F}$

$$
\begin{aligned}
\|g - f^*\|_{S'} - \|g - f^*\|_S &\leq \min_t \frac{\log M(t) - \log p}{t} \\
&\leq \min_{t, \epsilon} 2\epsilon + \frac{2B^2 t}{N} + \frac{\log \mathcal{N}(\mathcal{F}, \epsilon) - \log p}{t} \\
&= 2\min_\epsilon \epsilon + B\sqrt{2 \frac{\log \mathcal{N}(\mathcal{F}, \epsilon) - \log p}{N}}.
\end{aligned}
$$

$\square$

We can then use the above to bound the generalization error uniformly:

**Theorem 7.** *Under the same setup as Theorem 6, we have that with prob. $1 - p$ for all $g \in \mathcal{F}$*

$$
\|g - f^*\|_\pi - \|g - f^*\|_S \leq \min_\epsilon 2\epsilon + B\sqrt{8 \frac{\log \mathcal{N}(\mathcal{F}, \epsilon)}{N}} + cB\sqrt{\frac{-\log p/2}{N}},
$$

*for $c = \sqrt{2}(3 + \sqrt{5})$.*

*Proof.* Our goal is to approximate the population error $\|f^* - g\|_\pi^2$ by the test error $\|f^* - g\|_{S'}^2$ for a sample $S'$ independent of $g$, and this test error is itself close to the train error thanks to Theorem 6.

We use Bennett's inequality which relies on the fact that the function $h(x) = \|f^*(x) - g(x)\|^2$ is bounded by $4B^2$:

$$
\begin{aligned}
\mathbb{P}\left[\|f^* - g\|_\pi^2 - \|f^* - g\|_{S'}^2 \geq a\right] &= \mathbb{P}\left[\sum_{i=1}^N h(x_i) - N\mathbb{E}h(x) \geq Na\right] \\
&\leq \exp\left(-\frac{N\mathbb{E}_\pi\left[(h(x) - \mathbb{E}h(x))^2\right]}{(4B^2)^2}\phi\left(\frac{4B^2 a}{\mathbb{E}_\pi\left[(h(x) - \mathbb{E}h(x))^2\right]}\right)\right)
\end{aligned}
$$

for $\phi(u) = (1 + u)\log(1 + u) - u$.

Lemma 8 gives us the bound $\phi^{-1}(v) \leq \sqrt{2v} + 2v$, which implies that with probability $p$

$$
\begin{aligned}
\|f^* - g\|_\pi^2 - \|f^* - g\|_{S'}^2 &\leq \frac{\mathbb{E}_\pi\left[(h(x) - \mathbb{E}h(x))^2\right]}{4B^2}\left(\sqrt{\frac{-2(4B^2)^2 \log p}{N\mathbb{E}_\pi\left[(h(x) - \mathbb{E}h(x))^2\right]}} + \frac{-2(4B^2)^2 \log p}{N\mathbb{E}_\pi\left[(h(x) - \mathbb{E}h(x))^2\right]}\right) \\
&= \sqrt{2\mathbb{E}_\pi\left[(h(x) - \mathbb{E}h(x))^2\right]\frac{-\log p}{N}} + 8B^2 \frac{-\log p}{N}.
\end{aligned}
$$

We now use the facts that $\mathbb{E}_\pi\left[(h(x) - \mathbb{E}h(x))^2\right] \leq \mathbb{E}_\pi\left[h(x)^2\right] \leq 4B^2\mathbb{E}_\pi\left[h(x)\right] = 4B^2\|g - f^*\|_\pi^2$ to obtain

$$
\|f^* - g\|_\pi^2 - \|f^* - g\|_{S'}^2 \leq B\|g - f^*\|_\pi\sqrt{8\frac{-\log p}{N}} + 8B^2\frac{-\log p}{N}.
$$

Reordering this is implies

$$
\|f^* - g\|_\pi^2 - \|f^* - g\|_\pi\sqrt{8B^2\frac{-\log p}{N}} - \|f^* - g\|_{S'}^2 - 8B^2\frac{-\log p}{N} \leq 0
$$

which by the quadratic formula yields the inequality

$$
\begin{aligned}
\|f^* - g\|_\pi &\leq \sqrt{2B^2\frac{-\log p}{N}} + \sqrt{2B^2\frac{-\log p}{N} + \|f^* - g\|_{S'}^2 + 8B^2\frac{-\log p}{N}} \\
&\leq \|f^* - g\|_{S'} + (\sqrt{2} + \sqrt{10})B\sqrt{\frac{-\log p}{N}}
\end{aligned}
$$

This implies that with probability $1 - 2p$ we have that for all $g \in \mathcal{F}$,

$$\|g - f^*\|_\pi - \|g - f^*\|_S \le \|f^* - g\|_{S'} - \|g - f^*\|_S + (\sqrt{2} + \sqrt{10})B\sqrt{\frac{-\log p}{N}}$$

$$\le \min_\epsilon 2\epsilon + 2B\sqrt{2\frac{\log \mathcal{N}(\mathcal{F}, \epsilon) - \log p}{N}} + (\sqrt{2} + \sqrt{10})B\sqrt{\frac{-\log p}{N}}$$

$$\le \min_\epsilon 2\epsilon + B\sqrt{8\frac{\log \mathcal{N}(\mathcal{F}, \epsilon)}{N}} + \sqrt{2}(3 + \sqrt{5})B\sqrt{\frac{-\log p}{N}}.$$

$\square$

Let us now prove the lemma we used in the proof:

**Lemma 8.** *The function* $\phi(u) = (1 + u)\log(1 + u) - u$ *satisfies*

$$\phi(u) \le \frac{1}{1 + u}\frac{u^2}{2}.$$

*It is also invertible over the non-negative reals and its inverse satisfies*

$$\phi^{-1}(v) \le \sqrt{2v} + 2v.$$

*Proof.* To obtain the first inequality, we compute the derivatives of $\phi$:

$$\phi'(u) = \log(1 + u) + 1 - 1 = \log(1 + u)$$

$$\phi''(u) = \frac{1}{1 + u}.$$

which implies that

$$\phi(u) = \int_0^u \int_0^v \frac{1}{1 + w}dw\,dv \ge \frac{1}{1 + u}\int_0^u \int_0^v dw\,dv = \frac{1}{1 + u}\frac{u^2}{2}.$$

Solving for a non-negative $u$ in

$$v = \frac{1}{1 + u}\frac{u^2}{2}$$

yields

$$2v = \frac{1}{u^{-2} + u^{-1}}$$

$$\implies u^{-2} + u^{-1} - \frac{1}{2v} = 0$$

$$\implies u^{-1} = \frac{-1 + \sqrt{1 + \frac{2}{v}}}{2}$$

$$\implies u = \frac{2\sqrt{v}}{\sqrt{2 + v} - \sqrt{v}} = \sqrt{v}(\sqrt{2 + v} + \sqrt{v})$$

which implies that

$$\phi^{-1}(v) \le \sqrt{v}(\sqrt{2 + v} + \sqrt{v}) \le \sqrt{2v} + 2v.$$

$\square$

*Example* 1. As an example consider a function space $\mathcal{F}$ whose log covering number (this is also called the metric entropy) grows polynomially:

$$\log \mathcal{N}_2(\mathcal{F}, \epsilon) \le c\epsilon^{-r}.$$

Therefore we get that

$$\|f^* - f\|_\pi - \|f^* - f\|_S \le \min_\epsilon 2\epsilon + \sqrt{8}B\frac{\sqrt{c}}{\sqrt{\epsilon^r N}} + c_1 B\sqrt{\frac{-\log p/2}{N}}.$$

Choosing $\epsilon = \left(2B^2 \frac{c}{N}\right)^{\frac{1}{2+r}}$ (chosen so that the two terms depending on $\epsilon$ match), we obtain

$$\|f^* - f\|_\pi - \|f^* - f\|_S \le 2\left(2B^2 \frac{c}{N}\right)^{\frac{1}{2+r}} + c_1 B \sqrt{\frac{-\log p/2}{N}},$$

which is of order $N^{-\frac{1}{2+r}}$.

Assuming that the train error is zero (or sufficiently small, i.e. $O(N^{-\frac{1}{2+r}})$), we get that

$$\|f^* - f\|_\pi^2 = O(N^{-\frac{2}{2+r}}).$$

## C    ACCNET GENERALIZATION BOUNDS

The proof of generalization for shallow networks (Theorem 1) is the special case $L = 1$ of the proof of Theorem 2, so we only prove the second:

**Theorem 9.** *Consider an accordion net of depth $L$ and widths $d_L, \ldots, d_0$, with corresponding set of functions $\mathcal{F} = \{\sigma_L \circ f_{L:1} : \|f_\ell\|_{F_1} \le R_\ell, Lip(f_\ell) \le \rho_\ell\}$ with input space $\Omega = B(0, r)$ and with a final nonlinearity $\sigma_L$ that is $1$-Lipschitz and uniformly bounded by $B$. Then with probability $1 - p$ over the sampling of the training set $X$ from the distribution $\pi$, we have for all $f \in \mathcal{F}$*

$$\sqrt{\mathcal{L}(f)} - \sqrt{\tilde{\mathcal{L}}_N(f)} \le c_0 \sqrt{B\rho_{L:1} r \sum_{\ell'=1}^L \frac{R_{\ell'}}{\rho_{\ell'}} \sqrt{d_{\ell'} + d_{\ell'-1}} N^{-\frac{1}{2}} (1 + o(1))} + c_1 B \sqrt{\frac{-\log p/2}{N}},$$

*for $c_0 \approx 14.6$ and $c_1 \approx 7.4$. Thus if $\sqrt{\tilde{\mathcal{L}}_N(f)} \le c_0 \sqrt{B\rho_{L:1} r \sum_{\ell'=1}^L \frac{R_{\ell'}}{\rho_{\ell'}} \sqrt{d_{\ell'} + d_{\ell'-1}} N^{-\frac{1}{2}}} + c_1 B \sqrt{\frac{-\log p/2}{N}}$ we have*

$$\mathcal{L}(f) \le 8c_0^2 B\rho_{L:1} r \sum_{\ell'=1}^L \frac{R_{\ell'}}{\rho_{\ell'}} \sqrt{d_{\ell'} + d_{\ell'-1}} N^{-\frac{1}{2}} (1 + o(1)) + 8c_1^2 B^2 \frac{-\log p/2}{N}.$$

*Proof.* The strategy is: (1) prove a covering number bound on $\mathcal{F}$, (2) use it to bound the generalization error.

(1) We define $f_\ell = V_\ell \circ \sigma \circ W_\ell$ so that $f_\theta = f_{L:1} = f_L \circ \cdots \circ f_1$. First notice that we can write each $f_\ell$ as convex combination of its neurons:

$$f_\ell(x) = \sum_{i=1}^{w_\ell} v_{\ell,i} \sigma(w_{\ell,i}^T x) = R_\ell \sum_{i=1}^{w_\ell} c_{\ell,i} \bar{v}_{\ell,i} \sigma(\bar{w}_{\ell,i}^T x)$$

for $\bar{w}_{\ell,i} = \frac{w_{\ell,i}}{\|w_{\ell,i}\|}$, $\bar{v}_{\ell,i} = \frac{v_{\ell,i}}{\|v_{\ell,i}\|}$, $R_\ell = \sum_{i=1}^\ell \|v_{\ell,i}\| \|w_{\ell,i}\|$ and $c_{\ell,i} = \frac{1}{R_\ell} \|v_{\ell,i}\| \|w_{\ell,i}\|$.

Let us now consider a sequence $\epsilon_k = 2^{-k}$ for $k = 0, \ldots, K$ and define $\tilde{v}_{\ell,i}^{(k)}, \tilde{w}_{\ell,i}^{(k)}$ to be the $\epsilon_k$-covers of $\bar{v}_{\ell,i}, \bar{w}_{\ell,i}$, furthermore we may choose $\tilde{v}_{\ell,i}^{(0)} = \tilde{w}_{\ell,i}^{(0)} = 0$ since every unit vector is within a $\epsilon_0 = 1$ distance of the origin. We will now show that one can approximate $f_\theta$ by approximating each of the $f_\ell$ by functions of the form

$$\tilde{f}_\ell(x) = R_\ell \sum_{k=1}^{K_\ell} \frac{1}{M_{k,\ell}} \sum_{m=1}^{M_{k,\ell}} \tilde{v}_{\ell,i_{\ell,m}^{(k)}}^{(k)} \sigma(\tilde{w}_{\ell,i_{\ell,m}^{(k)}}^{(k)T} x) - \tilde{v}_{\ell,i_{\ell,m}^{(k)}}^{(k-1)} \sigma(\tilde{w}_{\ell,i_{\ell,m}^{(k)}}^{(k-1)T} x)$$

for indices $i_{\ell,m}^{(k)} = 1, \ldots, w_\ell$ choosen adequately. Notice that the number of functions of this type equals the number of $M_{k,\ell}$ quadruples $(\tilde{v}_{\ell,i_{\ell,m}^{(k)}}^{(k)}, \tilde{w}_{\ell,i_{\ell,m}^{(k)}}^{(k)T}, \tilde{v}_{\ell,i_{\ell,m}^{(k)}}^{(k-1)}, \tilde{w}_{\ell,i_{\ell,m}^{(k)}}^{(k-1)T})$ where these vectors belong to the $\epsilon_k$- resp. $\epsilon_{k-1}$-coverings of the $d_{in}$- resp. $d_{out}$-dimensional unit sphere. Thus the number of such functions is bounded by

$$\prod_{k=1}^{K_\ell} \left(\mathcal{N}_2(\mathbb{S}^{d_{in}-1}, \epsilon_k) \mathcal{N}_2(\mathbb{S}^{d_{out}-1}, \epsilon_k) \mathcal{N}_2(\mathbb{S}^{d_{in}-1}, \epsilon_{k-1}) \mathcal{N}_2(\mathbb{S}^{d_{out}-1}, \epsilon_{k-1})\right)^{M_{k,\ell}},$$

and we have this choice for all $\ell = 1, \ldots, L$. We will show that with sufficiently large $M_{k,\ell}$ this set of functions $\epsilon$-covers $\mathcal{F}$ which then implies that

$$\log \mathcal{N}_2(\mathcal{F}, \epsilon) \leq 2 \sum_{\ell=1}^{L} \sum_{k=1}^{K_\ell} M_{k,\ell} \left( \log \mathcal{N}_2(\mathbb{S}^{d_{in}-1}, \epsilon_{k-1}) + \log \mathcal{N}_2(\mathbb{S}^{d_{in}-1}, \epsilon_{k-1}) \right).$$

We will use the probabilistic method to find the right indices $i_{\ell,m}^{(k)}$ to approximate a function $f_\ell = R_\ell \sum_{i=1}^{w_\ell} c_{\ell,i} \bar{v}_{\ell,i} \sigma(\bar{w}_{\ell,i}^T x)$ with a function $\tilde{f}_\ell$. We take all $i_{\ell,m}^{(k)}$ to be i.i.d. equal to the index $i = 1, \cdots, w_\ell$ with probability $c_{\ell,i}$, so that in expectation

$$\mathbb{E} \tilde{f}_\ell(x) = R_\ell \sum_{k=1}^{K_\ell} \sum_{i=1}^{w_\ell} c_{\ell,i} \left( \tilde{v}_{\ell,i}^{(k)} \sigma(\tilde{w}_{\ell,i}^{(k)T} x) - \tilde{v}_{\ell,i}^{(k-1)} \sigma(\tilde{w}_{\ell,i}^{(k-1)T} x) \right)$$

$$= R_\ell \sum_{i=1}^{w_\ell} c_{\ell,i} \tilde{v}_{\ell,i}^{(K)} \sigma(\tilde{w}_{\ell,i}^{(K)T} x).$$

We will show that this expectation is $O(\epsilon_{K_\ell})$-close to $f_\ell$ and that the variance of $\tilde{f}_\ell$ goes to zero as the $M_{\ell,k}$s grow, allowing us to bound the expected error $\mathbb{E} \left\| f_{L:1} - \tilde{f}_{L:1} \right\|_\pi^2 \leq \epsilon^2$ which then implies that there must be at least one choice of indices $i_{\ell,m}^{(k)}$ such that $\left\| f_{L:1} - \tilde{f}_{L:1} \right\|_\pi \leq \epsilon$.

Let us first bound the distance

$$\left\| f_\ell(x) - \mathbb{E} \tilde{f}_\ell(x) \right\| = R_\ell \left\| \sum_{i=1}^{w_\ell} c_{\ell,i} \left( \bar{v}_{\ell,i} \sigma(\bar{w}_{\ell,i}^T x) - \tilde{v}_{\ell,i}^{(K)} \sigma(\tilde{w}_{\ell,i}^{(K)T} x) \right) \right\|$$

$$\leq R_\ell \sum_{i=1}^{w_\ell} c_{\ell,i} \left( \left\| \left( \bar{v}_{\ell,i} - \tilde{v}_{\ell,i}^{(K)} \right) \sigma(\bar{w}_{\ell,i}^T x) \right\| + \left\| \tilde{v}_{\ell,i}^{(K)} \left( \sigma(\bar{w}_{\ell,i}^T x) - \sigma(\tilde{w}_{\ell,i}^{(K)T} x) \right) \right\| \right)$$

$$\leq R_\ell \sum_{i=1}^{w_\ell} c_{\ell,i} \left( \left\| \bar{v}_{\ell,i} - \tilde{v}_{\ell,i}^{(K)} \right\| \left\| \bar{w}_{\ell,i}^T x \right\| + \left\| \tilde{v}_{\ell,i}^{(K)} \right\| \left\| \bar{w}_{\ell,i}^T x - \tilde{w}_{\ell,i}^{(K)T} x \right\| \right)$$

$$\leq 2R_\ell \sum_{i=1}^{w_\ell} c_{\ell,i} \epsilon_{K_\ell} \|x\|$$

$$= 2R_\ell \epsilon_{K_\ell} \|x\|.$$

Then we bound the trace of the covariance of $\tilde{f}_\ell$ which equals the expected square distance between $\tilde{f}_\ell$ and its expectation:

$$
\mathbb{E}\left\|\tilde{f}_\ell(x) - \mathbb{E}\tilde{f}_\ell(x)\right\|^2
$$

$$
= \sum_{k=1}^{K_\ell} \frac{R_\ell^2}{M_{k,\ell}^2} \sum_{m=1}^{M_{k,\ell}} \mathbb{E}\left\|\tilde{v}_{\ell,i_{\ell,m}^{(k)}}^{(k)} \sigma(\tilde{w}_{\ell,i_{\ell,m}^{(k)}}^{(k)T} x) - \tilde{v}_{\ell,i_{\ell,m}^{(k)}}^{(k-1)} \sigma(\tilde{w}_{\ell,i_{\ell,m}^{(k)}}^{(k-1)T} x) - \mathbb{E}\left[\tilde{v}_{\ell,i_{\ell,m}^{(k)}}^{(k)} \sigma(\tilde{w}_{\ell,i_{\ell,m}^{(k)}}^{(k)T} x) - \tilde{v}_{\ell,i_{\ell,m}^{(k)}}^{(k-1)} \sigma(\tilde{w}_{\ell,i_{\ell,m}^{(k)}}^{(k-1)T} x)\right]\right\|^2
$$

$$
\leq \sum_{k=1}^{K_\ell} \frac{R_\ell^2}{M_{k,\ell}^2} \sum_{m=1}^{M_{k,\ell}} \mathbb{E}\left\|\tilde{v}_{\ell,m}^{(k)} \sigma(\tilde{w}_{\ell,m}^{(k)T} x) - \tilde{v}_{\ell,m}^{(k-1)} \sigma(\tilde{w}_{\ell,m}^{(k-1)T} x)\right\|^2
$$

$$
= \sum_{k=1}^{K_\ell} \frac{R_\ell^2}{M_{k,\ell}} \sum_{i=1}^{w_\ell} c_i \left\|\tilde{v}_{\ell,i}^{(k)} \sigma\left(\tilde{w}_{\ell,i}^{(k)T} x\right) - \tilde{v}_{\ell,i}^{(k-1)} \sigma\left(\tilde{w}_{\ell,i}^{(k-1)T} x\right)\right\|^2
$$

$$
\leq \sum_{k=1}^{K_\ell} \frac{2R_\ell^2 \|x\|^2}{M_{k,\ell}} \sum_{i=1}^{w_\ell} c_i \left\|\tilde{v}_{\ell,i}^{(k)}\right\|^2 \left\|\tilde{w}_{\ell,i}^{(k)} - \tilde{w}_{\ell,i}^{(k-1)}\right\|^2 + c_i \left\|\tilde{v}_{\ell,i}^{(k)} - \tilde{v}_{\ell,i}^{(k-1)}\right\|^2 \left\|\tilde{w}_{\ell,i}^{(k-1)}\right\|^2
$$

$$
\leq \sum_{k=1}^{K_\ell} \frac{4R_\ell^2 \|x\|^2}{M_{k,\ell}} (\epsilon_k + \epsilon_{k-1})^2
$$

$$
\leq \sum_{k=1}^{K_\ell} \frac{36R_\ell^2 \|x\|^2}{M_{k,\ell}} \epsilon_k^2.
$$

Putting both together, we obtain

$$
\mathbb{E}\left\|f_\ell(x) - \tilde{f}_\ell(x)\right\|^2 \leq 4R_\ell^2 \epsilon_{K_\ell}^2 \|x\|^2 + \sum_{k=1}^{K_\ell} \frac{36R_\ell^2 \|x\|^2}{M_{k,\ell}} \epsilon_k^2
$$

$$
= 4R_\ell^2 \|x\|^2 \left(\epsilon_{K_\ell}^2 + 9\sum_{k=1}^{K_\ell} \frac{\epsilon_k^2}{M_{k,\ell}}\right).
$$

We will now use this bound, together with the Lipschitzness of $f_\ell$ to bound the error $\mathbb{E}\left\|f_{L:1}(x) - \tilde{f}_{L:1}(x)\right\|^2$. We will do this by induction on the distances $\mathbb{E}\left\|f_{\ell:1}(x) - \tilde{f}_{\ell:1}(x)\right\|^2$. We start by

$$
\mathbb{E}\left\|f_1(x) - \tilde{f}_1(x)\right\|^2 \leq 4R_1^2 \|x\|^2 \left(\epsilon_{K_\ell}^2 + 9\sum_{k=1}^{K_\ell} \frac{\epsilon_k^2}{M_{k,1}}\right).
$$

And for the induction step, we condition on the first $\ell - 1$ layers

$$
\mathbb{E}\left\|f_{\ell:1}(x) - \tilde{f}_{\ell:1}(x)\right\|^2 = \mathbb{E}\left[\mathbb{E}\left[\left\|f_{\ell:1}(x) - \tilde{f}_{\ell:1}(x)\right\|^2 | \tilde{f}_{\ell-1:1}\right]\right]
$$

$$
= \mathbb{E}\left\|f_{\ell:1}(x) - \mathbb{E}\left[\tilde{f}_{\ell:1}(x) | \tilde{f}_{\ell-1:1}\right]\right\|^2 + \mathbb{E}\mathbb{E}\left[\left\|\tilde{f}_{\ell:1}(x) - \mathbb{E}\left[\tilde{f}_{\ell:1}(x) | \tilde{f}_{\ell-1:1}\right]\right\|^2 | \tilde{f}_{\ell-1:1}\right]
$$

$$
= \mathbb{E}\left\|f_{\ell:1}(x) - f_\ell(\tilde{f}_{\ell-1:1}(x))\right\|^2 + \mathbb{E}\mathbb{E}\left[\left\|\tilde{f}_{\ell:1}(x) - f_\ell(\tilde{f}_{\ell-1:1}(x))\right\|^2 | \tilde{f}_{\ell-1:1}\right]
$$

$$
\leq \rho_\ell^2 \mathbb{E}\left\|f_{\ell-1:1}(x) - \tilde{f}_{\ell-1:1}(x)\right\|^2 + 4R_\ell^2 \mathbb{E}\left\|\tilde{f}_{\ell-1:1}(x)\right\|^2 \left(\epsilon_{K_\ell}^2 + 9\sum_{k=1}^{K_\ell} \frac{\epsilon_k^2}{M_{k,\ell}}\right).
$$

Now since

$$
\mathbb{E}\left\|\tilde{f}_{\ell-1:1}(x)\right\|^2 \leq \|f_{\ell-1:1}(x)\|^2 + \mathbb{E}\left\|f_{\ell-1:1}(x) - \tilde{f}_{\ell-1:1}(x)\right\|^2
$$

$$
\leq \rho_{\ell-1}^2 \cdots \rho_1^2 \|x\|^2 + \mathbb{E}\left\|f_{\ell-1:1}(x) - \tilde{f}_{\ell-1:1}(x)\right\|^2
$$

we obtain that

$$\mathbb{E}\left\|f_{\ell:1}(x) - \tilde{f}_{\ell:1}(x)\right\|^2 \leq \left(\rho_\ell^2 + 4R_\ell^2\left(\epsilon_{K_\ell}^2 + 9\sum_{k=1}^{K_\ell}\frac{\epsilon_k^2}{M_{k,\ell}}\right)\right)\mathbb{E}\left\|f_{\ell-1:1}(x) - \tilde{f}_{\ell-1:1}(x)\right\|^2$$
$$+ 4R_\ell^2\rho_{\ell-1}^2\cdots\rho_1^2\|x\|^2\left(\epsilon_{K_\ell}^2 + 9\sum_{k=1}^{K_\ell}\frac{\epsilon_k^2}{M_{k,\ell}}\right).$$

We define $\tilde{\rho}_\ell^2 = \rho_\ell^2\left[1 + 4\frac{R_\ell^2}{\rho_\ell^2}\left(\epsilon_{K_\ell}^2 + 9\sum_{k=1}^{K_\ell}\frac{\epsilon_k^2}{M_{k,\ell}}\right)\right]$ and obtain

$$\mathbb{E}\left\|f_{L:1}(x) - \tilde{f}_{L:1}(x)\right\|^2 \leq 4\sum_{\ell=1}^{L}\tilde{\rho}_{L:\ell+1}^2 R_\ell^2\rho_{\ell-1:1}^2\|x\|^2\left(\epsilon_{K_\ell}^2 + 9\sum_{k=1}^{K_\ell}\frac{\epsilon_k^2}{M_{k,\ell}}\right).$$

Thus for any distribution $\pi$ over the ball $B(0, r)$, there is a choice of indices $i_{\ell,m}^{(k)}$ such that

$$\left\|f_{L:1} - \tilde{f}_{L:1}\right\|_\pi^2 \leq 4\sum_{\ell=1}^{L}\tilde{\rho}_{L:\ell+1}^2 R_\ell^2\rho_{\ell-1:1}^2 r^2\left(\epsilon_{K_\ell}^2 + 9\sum_{k=1}^{K_\ell}\frac{\epsilon_k^2}{M_{k,\ell}}\right).$$

We now simply need to choose $K_\ell$ and $M_{k,\ell}$ adequately. To reach an error of $2\epsilon$, we choose

$$K_\ell = \left\lceil -\log\epsilon + \frac{1}{2}\log\left[4\rho_{L:1}^2 r^2\left(\sum_{\ell'=1}^{L}\frac{R_{\ell'}}{\rho_{\ell'}}\sqrt{d_{\ell'} + d_{\ell'-1}}\right)\frac{R_\ell}{\rho_\ell\sqrt{d_\ell + d_{\ell-1}}}\right]\right\rceil$$

where $\rho_{L:1} = \prod_{\ell=1}^{L}\rho_\ell$. Notice that that $\epsilon_{K_\ell}^2 \leq \frac{1}{4\rho_{L:1}^2 r^2\left(\sum_{\ell'=1}^{L}\frac{R_{\ell'}}{\rho_{\ell'}}\sqrt{d_{\ell'} + d_{\ell'-1}}\right)}\frac{\rho_\ell\sqrt{d_\ell + d_{\ell-1}}}{R_\ell}\epsilon^2$.

Given $s_0 = \sum_{k=1}^{\infty}\sqrt{k}2^{-k} \approx 1.3473 < \infty$, we define

$$M_{k,\ell} = \left\lceil 36\rho_{L:1}^2 r^2 s_0\left(\sum_{\ell'=1}^{L}\frac{R_{\ell'}}{\rho_{\ell'}}\sqrt{d_{\ell'} + d_{\ell'-1}}\right)\frac{R_\ell}{\rho_\ell\sqrt{d_\ell + d_{\ell-1}}}\frac{2^{-k}}{\sqrt{k}}\frac{1}{\epsilon^2}\right\rceil.$$

So that for all $\ell$

$$4\frac{R_\ell^2}{\rho_\ell^2}\left(\epsilon_{K_\ell}^2 + 9\sum_{k=1}^{K_\ell}\frac{\epsilon_k^2}{M_{k,\ell}}\right) \leq \frac{\frac{R_\ell}{\rho_\ell}\sqrt{d_\ell + d_{\ell-1}}}{\rho_{L:1}^2 r^2\left(\sum_{\ell'=1}^{L}\frac{R_\ell}{\rho_\ell}\sqrt{d_\ell + d_{\ell-1}}\right)}\epsilon^2$$
$$+ \frac{\frac{R_\ell}{\rho_\ell}\sqrt{d_\ell + d_{\ell-1}}}{\rho_{L:1}^2 r^2\left(\sum_{\ell'=1}^{L}\frac{R_\ell}{\rho_\ell}\sqrt{d_\ell + d_{\ell-1}}\right)}\epsilon^2\frac{\sum_{k'=1}^{K_\ell}\sqrt{k'}2^{-k'}}{s_0}$$
$$\leq 2\frac{\frac{R_\ell}{\rho_\ell}\sqrt{d_\ell + d_{\ell-1}}}{\rho_{L:1}^2 r^2\left(\sum_{\ell'=1}^{L}\frac{R_\ell}{\rho_\ell}\sqrt{d_\ell + d_{\ell-1}}\right)}\epsilon^2.$$

Now this also implies that

$$\tilde{\rho}_\ell \leq \rho_\ell\exp\left(2\frac{\frac{R_\ell}{\rho_\ell}\sqrt{d_\ell + d_{\ell-1}}}{\rho_{L:1}^2 r^2\left(\sum_{\ell'=1}^{L}\frac{R_\ell}{\rho_\ell}\sqrt{d_\ell + d_{\ell-1}}\right)}\epsilon^2\right)$$

and thus

$$\tilde{\rho}_{L:\ell+1} \leq \rho_{L:\ell+1}\exp\left(2\frac{\sum_{\ell'=\ell+1}^{L}\frac{R_\ell}{\rho_\ell}\sqrt{d_\ell + d_{\ell-1}}}{\rho_{L:1}^2 r^2\left(\sum_{\ell'=1}^{L}\frac{R_\ell}{\rho_\ell}\sqrt{d_\ell + d_{\ell-1}}\right)}\epsilon^2\right) \leq \rho_{L:\ell+1}\exp\left(\frac{2}{\rho_{L:1}^2 r^2}\epsilon^2\right).$$

Putting it all together, we obtain that

$$\left\| f_{L:1} - \tilde{f}_{L:1} \right\|_\pi^2 \le 4 \sum_{\ell=1}^L \tilde{\rho}_{L:\ell+1}^2 R_\ell^2 \rho_{\ell-1:1}^2 r^2 \left( \epsilon_{K_\ell}^2 + 9 \sum_{k=1}^{K_\ell} \frac{\epsilon_k^2}{M_{k,\ell}} \right)$$

$$\le \exp\left( \frac{2}{\rho_{L:1}^2 r^2} \epsilon^2 \right) \rho_{L:1}^2 r^2 \sum_{\ell=1}^L 4 \frac{R_\ell^2}{\rho_\ell^2} \left( \epsilon_{K_\ell}^2 + 9 \sum_{k=1}^{K_\ell} \frac{\epsilon_k^2}{M_{k,\ell}} \right)$$

$$\le 2 \exp\left( \frac{2}{\rho_{L:1}^2 r^2} \epsilon^2 \right) \epsilon^2$$

$$= 2\epsilon^2 + O(\epsilon^4).$$

Now since $\log \mathcal{N}_2(\mathbb{S}^{d_\ell - 1}, \epsilon) = d_\ell \log\left( \frac{1}{\epsilon} + 1 \right)$ and

$$M_{k,\ell} \le 36 \rho_{L:1}^2 r^2 s_0 \left( \sum_{\ell'=1}^L \frac{R_{\ell'}}{\rho_{\ell'}} \sqrt{d_{\ell'} + d_{\ell'-1}} \right) \frac{R_\ell}{\rho_\ell \sqrt{d_\ell + d_{\ell-1}}} \frac{2^{-k}}{\sqrt{k}} \frac{1}{\epsilon^2} + 1,$$

we have

$$\log \mathcal{N}_2\left( \mathcal{F}, \sqrt{2} \exp\left( \frac{\epsilon^2}{\rho_{L:1}^2 r^2} \right) \epsilon \right) \le 2 \sum_{\ell=1}^L \sum_{k=1}^{K_\ell} M_{k,\ell} \left( \log \mathcal{N}_2(\mathbb{S}^{d_\ell-1}, \epsilon_{k-1}) + \log \mathcal{N}_2(\mathbb{S}^{d_{\ell-1}-1}, \epsilon_{k-1}) \right)$$

$$\le 2 \sum_{\ell=1}^L \sum_{k=1}^{K_\ell} M_{k,\ell} (d_\ell + d_{\ell-1}) \log\left( \frac{1}{\epsilon_{k-1}} + 1 \right)$$

$$\le 72 s_0 \rho_{L:1}^2 r^2 \left( \sum_{\ell'=1}^L \frac{R_{\ell'}}{\rho_{\ell'}} \sqrt{d_{\ell'} + d_{\ell'-1}} \right) \sum_{\ell=1}^L \frac{R_\ell}{\rho_\ell} \sqrt{d_\ell + d_{\ell-1}} \sum_{k=1}^{K_\ell} \frac{2^{-k} \log\left( \frac{1}{\epsilon_{k-1}} + 1 \right)}{\sqrt{k}} \frac{1}{\epsilon^2}$$

$$+ 2 \sum_{\ell=1}^L (d_\ell + d_{\ell-1}) \sum_{k=1}^{K_\ell} \log\left( \frac{1}{\epsilon_{k-1}} + 1 \right)$$

$$\le 72 s_0^2 \rho_{L:1}^2 r^2 \left( \sum_{\ell'=1}^L \frac{R_{\ell'}}{\rho_{\ell'}} \sqrt{d_{\ell'} + d_{\ell'-1}} \right)^2 \frac{1}{\epsilon^2} + o(\epsilon^{-2}).$$

The diameter of $\mathcal{F}$ is smaller than $\rho_{L:1} r$, so for all $\delta \ge \rho_{L:1} r$, $\log \mathcal{N}_2(\mathcal{F}, \delta) = 0$. For all $\delta \le \rho_{L:1} r$ we choose $\epsilon = \frac{\delta}{\sqrt{2e}}$ so that $\sqrt{2} \exp\left( \frac{\epsilon^2}{\rho_{L:1}^2 r^2} \right) \epsilon \le \delta$ and therefore

$$\log \mathcal{N}_2(\mathcal{F}, \delta) \le 144 s_0^2 e \rho_{L:1}^2 r^2 \left( \sum_{\ell'=1}^L \frac{R_{\ell'}}{\rho_{\ell'}} \sqrt{d_{\ell'} + d_{\ell'-1}} \right)^2 \frac{1}{\delta^2} + o(\delta^{-2}).$$

(2) We now apply Theorem 7 along the lines of Example 1 with $r = 2$ and

$$c = 144 s_0^2 e \rho_{L:1}^2 r^2 \left( \sum_{\ell'=1}^L \frac{R_{\ell'}}{\rho_{\ell'}} \sqrt{d_{\ell'} + d_{\ell'-1}} \right)^2 \frac{1}{\delta^2},$$

to obtain

$$\sqrt{\mathcal{L}(f)} - \sqrt{\tilde{\mathcal{L}}_N(f)} \le 2 \sqrt{24 B s_0 \sqrt{e} \rho_{L:1} r \sum_{\ell'=1}^L \frac{R_{\ell'}}{\rho_{\ell'}} \sqrt{d_{\ell'} + d_{\ell'-1}} N^{-\frac{1}{4}} (1 + o(1))}$$

$$+ c_1 B \sqrt{\frac{-\log p/2}{N}}$$

$$= c_0 \sqrt{B \rho_{L:1} r \sum_{\ell'=1}^L \frac{R_{\ell'}}{\rho_{\ell'}} \sqrt{d_{\ell'} + d_{\ell'-1}} N^{-\frac{1}{4}} (1 + o(1))} + c_1 B \sqrt{\frac{-\log p/2}{N}},$$

for $c_0 = 2\sqrt{24 s_0 \sqrt{e}} \approx 14.6$ and $c_1 = \sqrt{2}(3 + \sqrt{5}) \approx 7.4$.

Therefore if $\sqrt{\tilde{\mathcal{L}}_N(f)} \le c_0 \sqrt{B\rho_{L:1} r \sum_{\ell'=1}^{L} \frac{R_{\ell'}}{\rho_{\ell'}} \sqrt{d_{\ell'} + d_{\ell'-1}} N^{-\frac{1}{4}}} + c_1 B \sqrt{\frac{-\log p/2}{N}}$, we have

$$
\mathcal{L}(f) \le \left( 2c_0 \sqrt{B\rho_{L:1} r \sum_{\ell'=1}^{L} \frac{R_{\ell'}}{\rho_{\ell'}} \sqrt{d_{\ell'} + d_{\ell'-1}} N^{-\frac{1}{4}} (1 + o(1))} + 2c_1 B \sqrt{\frac{-\log p/2}{N}} \right)^2
$$

$$
\le 8 c_0^2 B \rho_{L:1} r \sum_{\ell'=1}^{L} \frac{R_{\ell'}}{\rho_{\ell'}} \sqrt{d_{\ell'} + d_{\ell'-1}} N^{-\frac{1}{2}} (1 + o(1)) + 8 c_1^2 B^2 \frac{-\log p/2}{N}.
$$

$\square$

## D COMPOSITION OF SOBOLEV BALLS

Let us first prove a simple generalization bound for Sobolev balls:

**Proposition 10** (Proposition 3 from the main.)**.** *Given a distribution $\pi$ with support in $B(0, b)$, we have that with probability $1 - p$ for all functions $f \in \mathcal{F} = \{f : \|f\|_{W^{\nu,2}} \le R, \|f\|_\infty \le B\}$*

$$
\sqrt{\mathcal{L}(f)} - \sqrt{\tilde{\mathcal{L}}_N(f)} = c_0 \left( \frac{B^2 R^r}{N} \right)^{\frac{1}{2+r}} + c_1 B \sqrt{\frac{-\log p/2}{N}},
$$

*where $r = \frac{d}{\nu}$. Therefore if $\sqrt{\tilde{\mathcal{L}}_N(f)} \le c_0 \left( \frac{B^2 R^r}{N} \right)^{\frac{1}{2+r}} + c_1 B \sqrt{\frac{-\log p/2}{N}}$, we have*

$$
\mathcal{L}(f) \le 8 c_0^2 \left( \frac{B^2 R^r}{N} \right)^{\frac{2}{2+r}} + 8 c_1^2 B^2 \frac{-\log p/2}{N}.
$$

*Proof.* (1) We know from Theorem 5.2 of (Birman & Solomjak, 1967) that the Sobolev ball $B_{W^{\nu,2}}(0, R)$ over any $d$-dimensional hypercube $\Omega$ satisfies

$$
\log \mathcal{N}_2(B_{W^{\nu,2}}(0, R), \epsilon) \le C_0 \left( \frac{R}{\epsilon} \right)^{\frac{d}{\nu}}
$$

for a constant $c$ and any measure $\pi$ supported in the hypercube.

(2) We now apply Theorem 7 along the lines of Example 1 with $r = \frac{d}{\nu}$ and $c = C_0 R^r$, to obtain that

$$
\sqrt{\mathcal{L}(f)} - \sqrt{\tilde{\mathcal{L}}_N(f)} \le 2 \left( 2B^2 \frac{C_0 R^r}{N} \right)^{\frac{1}{2+r}} + (\sqrt{2} + c_1 B \sqrt{\frac{-\log p/2}{N}}
$$

$$
= c_0 \left( \frac{B^2 R^r}{N} \right)^{\frac{1}{2+r}} + c_1 B \sqrt{\frac{-\log p/2}{N}}.
$$

$\square$

The bound on the covering number of Sobolev balls, can then be used to bound compositions of Sobolev balls, thanks to the following general result:

**Proposition 11.** *Let $\mathcal{F}_1, \ldots, \mathcal{F}_L$ be set of functions mapping through the sets $\Omega_0, \ldots, \Omega_L$, then if all functions in $\mathcal{F}_\ell$ are $\rho_\ell$-Lipschitz, we have*

$$
\log \mathcal{N}_2(\mathcal{F}_L \circ \cdots \circ \mathcal{F}_1, \sum_{\ell=1}^{L} \rho_{L:\ell+1} \epsilon_\ell) \le \sum_{\ell=1}^{L} \log \mathcal{N}_2(\mathcal{F}_\ell, \epsilon_\ell).
$$

*Proof.* For any distribution $\pi_0$ on $\Omega$ there is a $\epsilon_1$-covering $\tilde{\mathcal{F}}_1$ of $\mathcal{F}_1$ with $\left| \tilde{\mathcal{F}}_1 \right| \le \mathcal{N}_2(\mathcal{F}_1, \epsilon_1)$ then for any $\tilde{f}_1 \in \tilde{\mathcal{F}}_1$ we choose a $\epsilon_2$-covering $\tilde{\mathcal{F}}_2$ w.r.t. the measure $\pi_1$ which is the measure of $f_1(x)$ if

$x \sim \pi_0$ of $\mathcal{F}_2$ with $\left|\tilde{\mathcal{F}}_2\right| \leq \mathcal{N}_2(\mathcal{F}_2, \epsilon)$, and so on until we obtain coverings for all $\ell$. Then the set $\tilde{\mathcal{F}} = \left\{ \tilde{f}_L \circ \cdots \circ \tilde{f}_1 : \tilde{f}_1 \in \tilde{\mathcal{F}}_1, \ldots, \tilde{f}_L \in \tilde{\mathcal{F}}_L \right\}$ is a $\sum_{\ell=1}^{L} \rho_{L:\ell+1}\epsilon_\ell$-covering of $\mathcal{F} = \mathcal{F}_L \circ \cdots \circ \mathcal{F}_1$, indeed for any $f = f_{L:1}$ we choose $\tilde{f}_1 \in \tilde{\mathcal{F}}_1, \ldots, \tilde{f}_L \in \tilde{\mathcal{F}}_L$ that cover $f_1, \ldots, f_L$, then $\tilde{f}_{L:1}$ covers $f_{L:1}$:

$$\left\| f_{L:1} - \tilde{f}_{L:1} \right\|_\pi \leq \sum_{\ell=1}^{L} \left\| f_{L:\ell} \circ \tilde{f}_{\ell-1:1} - f_{L:\ell+1} \circ \tilde{f}_{\ell:1} \right\|_\pi$$

$$\leq \sum_{\ell=1}^{L} \left\| f_{L:\ell} - f_{L:\ell+1} \circ \tilde{f}_\ell \right\|_{\pi_{\ell-1}}$$

$$\leq \sum_{\ell=1}^{L} \rho_{L:\ell+1}\epsilon_\ell,$$

and log cardinality of the set $\tilde{\mathcal{F}}$ is bounded $\sum_{\ell=1}^{L} \log \mathcal{N}_2(\mathcal{F}_\ell, \epsilon_\ell)$. $\qquad\square$

We can now put the two previous results together to obtain a generalization bound for composition of Sobolev balls:

**Theorem 12.** *Let $\mathcal{F} = \mathcal{F}_L \circ \cdots \circ \mathcal{F}_1$ where $\mathcal{F}_\ell = \left\{ f_\ell : \mathbb{R}^{d_{\ell-1}} \to \mathbb{R}^{d_\ell} \text{ s.t. } \|f_\ell\|_{W^{\nu_\ell,2}} \leq R, \|f_\ell\|_\infty \leq B, Lip(f_\ell) \leq \rho_\ell \right\}$, and let $r^* = \min_\ell r_\ell$ for $r_\ell = \frac{\nu_\ell}{d_{\ell-1}}$, then with probability $1 - p$ we have for all $f \in \mathcal{F}$*

$$\sqrt{\mathcal{L}(f)} - \sqrt{\tilde{\mathcal{L}}_N(f)} \leq c_0 \left( \frac{B^2 R^{r^*}}{N} \right)^{\frac{1}{2+r^*}} + c_1 B \sqrt{\frac{-\log p/2}{N}},$$

*where $c_0, c_1$ depend only on $r_\ell, \rho_\ell$ and $L$. Thus if $\sqrt{\tilde{\mathcal{L}}_N(f)} \leq c_0 \left( \frac{B^2 R^{r^*}}{N} \right)^{\frac{1}{2+r^*}} + c_1 B \sqrt{\frac{-\log p/2}{N}}$, we have*

$$\mathcal{L}(f) \leq 8c_0^2 \left( \frac{B^2 R^{r^*}}{N} \right)^{\frac{2}{2+r^*}} + 8c_1^2 B^2 \frac{-\log p/2}{N}.$$

*Proof.* (1) We know from Theorem 5.2 of (Birman & Solomjak, 1967) that the Sobolev ball $B_{W^{\nu_\ell,2}}(0, R)$ over any $d_\ell$-dimensional hypercube $\Omega$ satisfies

$$\log \mathcal{N}_2(B_{W^{\nu,2}}(0, R), \pi_{\ell-1}, \epsilon_\ell) \leq \left( C_\ell \frac{R}{\epsilon_\ell} \right)^{r_\ell}$$

for $r_\ell = \frac{d_\ell}{\nu_\ell}$ and a constant $C_\ell$ that depends on the size of hypercube and the dimension $d_\ell$ and the regularity $\nu_\ell$ and any measure $\pi_{\ell-1}$ supported in the hypercube.

Thus Proposition 11 tells us that the composition of the Sobolev balls satisfies

$$\log \mathcal{N}_2(\mathcal{F}_L \circ \cdots \circ \mathcal{F}_1, \sum_{\ell=1}^{L} \rho_{L:\ell+1}\epsilon_\ell) \leq \sum_{\ell=1}^{L} \left( C_\ell \frac{R}{\epsilon_\ell} \right)^{r_\ell}.$$

Given $r^* = \max_\ell r_\ell$, we can bound it by $R^{r^*} \sum_{\ell=1}^{L} \left( C_\ell \frac{1}{\epsilon_\ell} \right)^{r^*}$ and by then choosing $\epsilon_\ell = \frac{\rho_{L:\ell+1}^{-1}(\rho_{L:\ell+1}C_\ell)^{\frac{r^*}{1+r^*}}}{\sum_\ell (\rho_{L:\ell+1}C_\ell)^{\frac{r^*}{1+r^*}}} \epsilon$, we obtain that

$$\log \mathcal{N}_2(\mathcal{F}_L \circ \cdots \circ \mathcal{F}_1, \epsilon) \leq \left( \sum_{\ell=1}^{L} (\rho_{L:\ell+1}C_\ell)^{\frac{r^*}{1+r^*}} \right)^{1+r^*} \left( \frac{R}{\epsilon} \right)^{r^*}.$$

(2) Again, we apply Theorem 7 along the lines of Example 1, we obtain that

$$\sqrt{\mathcal{L}(f)} - \sqrt{\tilde{\mathcal{L}}_N(f)} \leq c_0 \left( \frac{B^2 R^{r^*}}{N} \right)^{\frac{1}{2+r^*}} + c_1 B \sqrt{\frac{-\log p/2}{N}}.$$

$\qquad\square$

# E   GENERALIZATION AT THE REGULARIZED GLOBAL MINIMUM

In this section, we first give the proof of Theorem 5 and then present detailed proofs of lemmas used in the proof. The lemmas are largely inspired by (Bach, 2017) and may be of independent interest.

## E.1   THEOREM 5 IN SECTION 4.2

**Theorem 13** (Theorem 5 in the main). *Given a true function $f^*_{L^*:1} = f^*_{L^*} \circ \cdots \circ f^*_1$ going through the dimensions $d^*_0, \ldots, d^*_{L^*}$, along with a continuous input distribution $\pi_0$ supported in $B(0, b_0)$, such that the distributions $\pi_\ell$ of $f^*_\ell(x)$ (for $x \sim \pi_0$) are continuous too and supported inside $B(0, b_\ell) \subset \mathbb{R}^{d^*_\ell}$. Further assume that there are differentiabilities $\nu_\ell$ and radii $R_\ell$ such that $\|f^*_\ell\|_{W^{\nu_\ell,2}(B(0,b_\ell))} \leq R_\ell$, and $\rho_\ell$ such that $Lip(f^*_\ell) \leq \rho_\ell$. For an infinite width AccNet with $L \geq L^*$ and dimensions $d_\ell \geq d^*_1, \ldots, d^*_{L^*-1}$, we have for the ratios $\tilde{r}_\ell = \max\{\frac{d^*_{\ell-1}+3}{\nu_\ell}, 2\}$:*

*(1) At a global min $\hat{f}_{L:1}$ of $\tilde{\mathcal{L}}_N(f_{L:1}) + \lambda R(\theta)$, we have $\mathcal{L}(\hat{f}_{L:1}) = O(N^{-\frac{2}{2+\tilde{r}_{\max}}})$ for $\tilde{r}_{\max} = \max\{\tilde{r}_1, \ldots, \tilde{r}_L\}$.*

*(2) At a global min $\hat{f}_{L:1}$ of $\tilde{\mathcal{L}}_N(f_{L:1}) + \lambda\tilde{R}(\theta)$, we have $\mathcal{L}(\hat{f}_{L:1}) = O(N^{-\frac{2}{2+\tilde{r}_{sum}}})$ for $\tilde{r}_{sum} = 2 + \sum_\ell \tilde{r}_\ell - 2$.*

*Proof.* If $f^* = f^*_{L^*} \circ \cdots \circ f^*_1$ with $L^* \leq L$, intermediate dimensions $d^*_0, \ldots, d^*_{L^*}$, along with a continuous input distribution $\pi_0$ supported in $B(0, b_0)$, such that the distributions $\pi_\ell$ of $f^*_\ell(x)$ (for $x \sim \pi_0$) are continuous too and supported inside $B(0, b_\ell) \subset \mathbb{R}^{d^*_\ell}$. Further assume that there are differentiabilities $\nu^*_\ell$ and radii $R_\ell$ such that $\|f^*_\ell\|_{W^{\nu^*_\ell,2}(B(0,b_\ell))} \leq R_\ell$.

We first focus on the $L = L^*$ case and then extend to the $L > L^*$ case.

Each $f^*_\ell$ can be approximated by another function $\tilde{f}_\ell$ with bounded $F_1$-norm and Lipschitz constant. Actually if $2\nu_\ell \geq d^*_{\ell-1} + 3$ one can choose $\tilde{f}_\ell = f^*_\ell$ since $\|f^*_\ell\|_{F_1} \leq C_\ell R_\ell$ by Lemma 16, and by assumption $Lip(\tilde{f}_\ell) \leq \rho_\ell$. If $2\nu_\ell < d^*_{\ell-1} + 3$, then by Lemma 15 we know that there is a $\tilde{f}_\ell$ with

$$\left\|\tilde{f}_\ell\right\|_{F_1} \leq C_\ell R_\ell \epsilon_\ell^{-\frac{d^*_{\ell-1}+3}{2\nu_\ell}+1} \text{ and } Lip(\tilde{f}_\ell) \leq C_\ell Lip(f^*_\ell) \leq C_\ell \rho_\ell \text{ and error}$$

$$\left\|f^*_\ell - \tilde{f}_\ell\right\|_{L_2(\pi_{\ell-1})} \leq c_\ell \left\|f^* - \tilde{f}_\ell\right\|_{L_2(B(0,b_\ell))} \leq c_\ell \epsilon_\ell.$$

Note that by setting $\tilde{r}_\ell = \max\{\frac{d^*_{\ell-1}+3}{\nu_\ell}, 2\}$, we can write $\left\|\tilde{f}_\ell\right\|_{F_1} \leq C_\ell R_\ell \epsilon_\ell^{-\frac{\tilde{r}_\ell}{2}+1}$ in both cases.

Therefore the composition $\tilde{f}_{L:1}$ satisfies

$$\left\|f^*_{L:1} - \hat{f}_{L:1}\right\|_{L_2(\pi_{\ell-1})} \leq \sum_{\ell=1}^L \left\|\tilde{f}_{L:\ell+1} \circ f^*_{\ell:1} - \tilde{f}_{L:\ell} \circ f^*_{\ell-1:1}\right\|_{L_2(\pi)}$$

$$\leq \sum_{\ell=1}^L Lip(\tilde{f}_{L:\ell+1}) c_\ell \epsilon_\ell$$

$$\leq \sum_{\ell=1}^L \rho_{L:\ell+1} C_{L:\ell+1} c_\ell \epsilon_\ell.$$

For any $L \geq L^*$, dimensions $d_\ell \geq d^*_\ell$ and widths $w_\ell \geq N$, we can build an AccNet that fits exactly $\tilde{f}_{L:1}$, by simply adding zero weights along the additional dimensions and widths, and by adding identity layers if $L > L^*$, since it is possible to represent the identity on $\mathbb{R}^d$ with a shallow network with $2d$ neurons and $F_1$-norm $2d$ (by having two neurons $e_i \sigma(e_i^T \cdot)$ and $-e_i \sigma(-e_i^T \cdot)$ for each basis $e_i$). Since the cost in parameter norm of representing the identity scales with the dimension, it is best to add those identity layers at the minimal dimension $\min\{d^*_0, \ldots, d^*_{L^*}\}$. We therefore end up with a AccNet with $L - L^*$ identity layers (with $F_1$ norm $2\min\{d^*_0, \ldots, d^*_{L^*}\}$) and $L^*$ layers that approximate each of the $f^*_\ell$ with a bounded $F_1$-norm function $\tilde{f}_\ell$.

Since $f_{L:1}^*$ has zero population loss, the population loss of the AccNet $\tilde{f}_{L:1}$ is bounded by $(\rho \sum_{\ell=1}^{L} \rho_{L:\ell+1} C_{L:\ell+1} c_\ell \epsilon_\ell)^2$. Using Bennett's inequality as in the proof of Theorem 7 (though in the 'other direction' since we want to bound $\tilde{\mathcal{L}}_N$ in terms of $\mathcal{L}$ and not vice versa), we obtain that with probability $1 - p$

$$\tilde{\mathcal{L}}_N(\tilde{f}_{L:1}) \leq \mathcal{L}(\tilde{f}_{L:1}) + \sqrt{8B^2 \mathcal{L}(\tilde{f}_{L:1}) \frac{-\log p}{N}} + 8B^2 \frac{-\log p}{N}$$

$$\leq \left( \sqrt{\mathcal{L}(\tilde{f}_{L:1})} + \sqrt{8B^2 \frac{-\log p}{N}} \right)^2$$

$$\leq \left( \rho \sum_{\ell=1}^{L} \rho_{L:\ell+1} C_{L:\ell+1} c_\ell \epsilon_\ell + \sqrt{8B^2 \frac{-\log p}{N}} \right)^2$$

(1) At the global minimizer $\hat{f}_{L:1} = \hat{f}_L \circ \cdots \circ \hat{f}_1$ of the regularized loss (with the first regularization term), the regularized loss is upper bounded by

$$\left( \rho \sum_{\ell=1}^{L} \rho_{L:\ell+1} C_{L:\ell+1} c_\ell \epsilon_\ell + \sqrt{8B^2 \frac{-\log p}{N}} \right)^2$$
$$+ \lambda \sqrt{2d} \prod_{\ell=1}^{L^*} C_\ell \rho_\ell \left[ \sum_{\ell=1}^{L^*} \frac{R_\ell \epsilon_\ell^{\frac{2 - \tilde{r}_\ell}{2}}}{C_\ell \rho_\ell} + 2(L - L^*) \min\{d_0^*, \dots, d_{L^*}^*\} \right].$$

Taking $\epsilon_\ell = N^{-\frac{1}{2 + \tilde{r}_\ell}}$ and $\lambda = N^{-\frac{1}{2}}$, this is upper bounded by

$$\left( \rho \sum_{\ell=1}^{L} \rho_{L:\ell+1} C_{L:\ell+1} c_\ell N^{-\frac{1}{2 + \tilde{r}_\ell}} + \sqrt{8B^2 \frac{-\log p}{N}} \right)^2$$
$$+ \sqrt{2d} \prod_{\ell=1}^{L^*} C_\ell \rho_\ell \left[ \sum_{\ell=1}^{L^*} \frac{R_\ell}{C_\ell \rho_\ell} N^{-\frac{2}{2 + \tilde{r}_\ell}} + 2(L - L^*) \min\{d_0^*, \dots, d_{L^*}^*\} N^{-\frac{1}{2}} \right].$$

The above is of order $N^{-\frac{2}{2 + \tilde{r}_{\max}}}$ for $\tilde{r}_{\max} = \max\{\tilde{r}_\ell : \ell = 1, \dots, L\}$, which implies that at the global minimizer of the regularized loss, the (unregularized) train loss is of order $N^{-\frac{2}{2 + \tilde{r}_{\max}}}$ and the complexity measure $R(\hat{f}_1, \dots, \hat{f}_L)$ is of order $N^{\frac{1}{2} - \frac{2}{2 + \tilde{r}_{\max}}}$ which implies that the test error will be of order $N^{-\frac{2}{2 + \tilde{r}_{\max}}}$ as well.

(2) Let us now focus on the $\tilde{R}(\theta)$ regularizer instead. Taking the same approximation $\tilde{f}_{L:1}$, we see that the global minimum $\hat{f}_{L:1}$ of the $\tilde{R}$-regularized loss is upper bounded by

$$\left( \rho \sum_{\ell=1}^{L} \rho_{L:\ell+1} C_{L:\ell+1} c_\ell \epsilon_\ell + \sqrt{8B^2 \frac{-\log p}{N}} \right)^2$$
$$+ \lambda \sqrt{2d} \left( L^* + (L - L^*) \min\{d_0^*, \dots, d_{L^*}^*\} \right) \prod_{\ell=1}^{L^*} C_\ell R_\ell \epsilon_\ell^{\frac{2 - \tilde{r}_\ell}{2}}.$$

where we used the bound $\|W_\ell\|_{op} \|V_\ell\|_{op} \leq \|f_\ell\|_{F_1}$.

Choosing $\epsilon_\ell = N^{-\frac{2}{2 + \tilde{r}_{sum}}}$ for $\tilde{r}_{sum} = 2 + \sum_\ell (\tilde{r}_\ell - 2)$ and $\lambda = N^{-\frac{1}{N}}$ is upper bounded by

$$\left( \rho \sum_{\ell=1}^{L} \rho_{L:\ell+1} C_{L:\ell+1} c_\ell N^{-\frac{1}{2 + \tilde{r}_{sum}}} + \sqrt{8B^2 \frac{-\log p}{N}} \right)^2$$
$$+ \sqrt{2d} \left( L^* + (L - L^*) \min\{d_0^*, \dots, d_{L^*}^*\} \right) \left[ \prod_{\ell=1}^{L^*} C_\ell R_\ell \right] N^{-\frac{2}{2 + \tilde{r}_{sum}}}.$$

Which implies that both the train error is of order $N^{-\frac{2}{2+\tilde{r}_{sum}}}$ and the regularization term is of order $N^{\frac{1}{2}-\frac{2}{2+\tilde{r}_{sum}}}$.

And since the $\tilde{R}$-regularized loss bounds the $R$-regularized loss, the test error will be of order $N^{-\frac{2}{2+\tilde{r}_{sum}}}$.

Note that if there is at a most one $\ell$ where $\tilde{r}_\ell > 2$ then the rate is the same for both regularizers. $\qquad\square$

## E.2 LEMMAS ON APPROXIMATING SOBOLEV FUNCTIONS

Now we present the lemmas used in this proof above that concern the approximation errors and Lipschitz constants of Sobolev functions and compositions of them. We will bound the $F_2$-norm and note that the $F_2$-norm is larger than the $F_1$-norm, cf. (Bach, 2017, Section 3.1).

**Lemma 14** (Approximation for Sobolev function with bounded error and Lipschitz constant). *Suppose $g : \mathbb{S}_d \to \mathbb{R}$ is an even function with bounded Sobolev norm $\|g\|_{W^{\nu,2}} \leq R$ with $2\nu \leq d+2$, with inputs on the unit $d$-dimensional sphere. Then for every $\epsilon > 0$, there is $\hat{g} \in \mathcal{G}_2$ with small approximation error $\|g - \hat{g}\|_{L_2(\mathbb{S}_d)} = C(d,\nu)R\epsilon$, bounded Lipschitzness $\mathrm{Lip}(\hat{g}) \leq C'(d)\mathrm{Lip}(g)$, and bounded norm*

$$\|\hat{g}\|_{F_2} \leq C''(d,\nu)R\epsilon^{-\frac{d+3-2\nu}{2\nu}}.$$

*Proof.* Given our assumptions on the target function $g$, we may decompose $g(x) = \sum_{k=0}^{\infty} g_k(x)$ along the basis of spherical harmonics with $g_0(x) = \int_{\mathbb{S}_d} g(y)\mathrm{d}\tau_d(y)$ being the mean of $g(x)$ over the uniform distribution $\tau_d$ over $\mathbb{S}_d$. The $k$-th component can be written as

$$g_k(x) = N(d,k)\int_{\mathbb{S}_d} g(y)P_k(x^T y)\mathrm{d}\tau_d(y)$$

with $N(d,k) = \frac{2k+d-1}{k}\binom{k+d-2}{d-1}$ and a Gegenbauer polynomial of degree $k$ and dimension $d+1$:

$$P_k(t) = (-1/2)^k \frac{\Gamma(d/2)}{\Gamma(k+d/2)}(1-t^2)^{(2-d)/2}\frac{d^k}{dt^k}(1-t^2)^{k+(d-2)/2},$$

known as Rodrigues' formula. Given the assumption that the Sobolev norm $\|g\|_{W^{\nu,2}}^2$ is upper bounded, we have $\|f\|_{L_2(\mathbb{S}_d)}^2 \leq C_0(d,\nu)R$ for $f = \Delta^{\nu/2}g$ where $\Delta$ is the Laplacian on $\mathbb{S}_d$ (Evans, 2022; Bach, 2017). Note that $g_k$ are eigenfunctions of the Laplacian with eigenvalues $k(k+d-1)$ (Atkinson & Han, 2012), thus

$$\|g_k\|_{L_2(\mathbb{S}_d)}^2 = \|f_k\|_{L_2(\mathbb{S}_d)}^2 (k(k+d-1))^{-\nu} \leq \|f_k\|_{L_2(\mathbb{S}_d)}^2 k^{-2\nu} \leq C_0(d,\nu)R^2 k^{-2\nu} \qquad (1)$$

where in the last inequality holds we use $\|f\|_{L_2(\mathbb{S}_d)}^2 = \sum_{k\geq 0}\|f_k\|_{L_2(\mathbb{S}_d)}^2$. Note using the Hecke-Funk formula, we can also write $g_k$ as scaled $p_k$ for the underlying density $p$ of the $F_1$ and $F_2$-norms:

$$g_k(x) = \lambda_k p_k(x)$$

where $\lambda_k = \frac{\omega_{d-1}}{\omega_d}\int_{-1}^1 \sigma(t)P_k(t)(1-t^2)^{(d-2)/2}\mathrm{d}t = \Omega(k^{-(d+3)/2})$ (Bach, 2017, Appendix D.2) and $\omega_d$ denotes the surface area of $\mathbb{S}_d$. Then by definition of $\|\cdot\|_{F_2}$, for some probability density $p$,

$$\|g\|_{F_2}^2 = \int_{\mathbb{S}_d}|p|^2\mathrm{d}\tau(v) = \|p\|_{L_2(\mathbb{S}_d)}^2 = \sum_{0\leq k}\|p_k\|_{L_2(\mathbb{S}_d)}^2 = \sum_{0\leq k}\lambda_k^{-2}\|g_k\|_{L_2(\mathbb{S}_d)}^2.$$

Now to approximate $g$, consider function $\hat{g}$ defined by truncating the "high frequencies" of $g$, i.e. setting $\hat{g}_k = \mathbb{1}[k \leq m]g_k$ for some $m > 0$ we specify later. Then we can bound the norm with

$$\|\hat{g}\|_{F_2}^2 = \sum_{0\leq k:\lambda_k\neq 0}\lambda_k^{-2}\|\hat{g}_k\|_{L_2(\mathbb{S}_d)}^2 = \sum_{\substack{0\leq k\leq m \\ \lambda_k\neq 0}}\lambda_k^{-2}\|g_k\|_{L_2(\mathbb{S}_d)}^2$$

$$\overset{(a)}{\leq} C_1(d,\nu)\sum_{0\leq k\leq m}\|f_k\|_{L_2(\mathbb{S}_d)}^2 k^{d+3-2\nu}$$

$$\leq C_1(d,\nu)m^{d+3-2\nu}\sum_{0\leq k\leq m}\|f_k\|_{L_2(\mathbb{S}_d)}^2$$

$$\leq C_2(d,\nu)R^2 m^{d+3-2\nu}$$

where (a) uses Eq 1 and $\lambda_k = \Omega(k^{-(d+3)/2})$.

To bound the approximation error,

$$
\begin{aligned}
\|g - \hat{g}\|^2_{L_2(\mathbb{S}_d)} = \left\|\sum_{k>m} g_k\right\|^2_{L_2(\mathbb{S}_d)} &\leq \sum_{k>m} \|g_k\|^2_{L_2(\mathbb{S}_d)} \\
&\leq \sum_{k>m} \|f_k\|^2_{L_2(\mathbb{S}_d)} k^{-2\nu} \\
&\leq C_0(d,\nu) R^2 m^{-2\nu} \quad \text{since } \sum_{k>m} \|f_k\|^2_{L_2(\mathbb{S}_d)} \leq \|f\|^2_{L_2(\mathbb{S}_d)}.
\end{aligned}
$$

Finally, choosing $m = \epsilon^{-\frac{1}{\nu}}$, we obtain $\|g - \hat{g}\|_{L_2(\mathbb{S}_d)} \leq C(d,\nu)R\epsilon$ and

$$
\|\hat{g}\|_{F_2} \leq C'(d,\nu) R\epsilon^{-\frac{d+3-2\nu}{2\nu}}.
$$

Then it remains to bound $\mathrm{Lip}(\hat{g})$ for our constructed approximation. By construction and by (Dai, 2013, Theorem 2.1.3), we have $\hat{g} = g * h$ with now

$$
h(t) = \sum_{k=0}^{m} h_k P_k(t), \quad t \in [-1, 1]
$$

by orthogonality of the Gegenbauer polynomial $P_k$'s and the convolution is defined as

$$
(g * h)(x) := \frac{1}{\omega_d} \int_{\mathbb{S}_d} g(y) h(\langle x, y \rangle) \mathrm{d}y.
$$

The coefficients for $0 \leq k \leq m$ given by (Dai, 2013, Theorem 2.1.3) are

$$
h_k \overset{(a)}{=} \frac{\omega_{d+1}}{\omega_d} \frac{\Gamma(d-1)}{\Gamma(d-1+k)} P_k(1) \frac{k!(k+(d-1)/2)\Gamma((d-1)/2)^2}{\pi 2^{2-d}\Gamma(d-1+k)} \overset{(b)}{=} O\left(\frac{k}{\Gamma(d-1+k)}\right)
$$

where (a) follows from the (inverse of) weighted $L_2$ norm of $P_k$; (b) plugs in the unit constant $P_k(1) = \frac{\Gamma(k+d-1)}{\Gamma(d-1)k!}$ and suppresses the dependence on $d$. Note that the constant factor $\frac{\Gamma(d-1)}{\Gamma(d-1+k)}$ comes from the difference in the definitions of the Gegenbauer polynomials here and in (Dai, 2013). Then we can bound

$$
\begin{aligned}
\|\nabla \hat{g}(x)\|_{op} &\leq \int_{\mathbb{S}_d} \|\nabla g(y)\|_{op} |h(\langle x, y \rangle)| \mathrm{d}y \\
&\leq \mathrm{Lip}(g) \int_{\mathbb{S}_d} |h(\langle x, y \rangle)| \mathrm{d}y \\
&\leq \sqrt{\omega_d} \mathrm{Lip}(g) \left(\int_{\mathbb{S}_d} h(\langle x, y \rangle)^2 \mathrm{d}y\right)^{1/2} \quad \text{by Cauchy-Schwartz} \\
&= \sqrt{\omega_d} \mathrm{Lip}(g) \left(\sum_{k,j=0}^{m} \int_{\mathbb{S}_d} h_k h_j P_k(\langle x, y \rangle) P_j(\langle x, y \rangle) \mathrm{d}y\right)^{1/2} \\
&= \sqrt{\omega_d} \mathrm{Lip}(g) \left(\sum_{k,j=0}^{m} \int_{-1}^{1} h_k h_j P_k(t) P_j(t) (1-t^2)^{\frac{d-2}{2}} \mathrm{d}t\right)^{1/2} \quad \text{by (Dai, 2013, Eq A.5.1)} \\
&= \sqrt{\omega_d} \mathrm{Lip}(g) \left(\sum_{k=0}^{m} h_k^2 \int_{-1}^{1} P_k(t)^2 (1-t^2)^{\frac{d-2}{2}} \mathrm{d}t\right)^{1/2} \quad \text{by orthogonality of } P_k\text{'s w.r.t. this measure} \\
&= \sqrt{\omega_d} \mathrm{Lip}(g) \left(\sum_{k=0}^{m} h_k^2 \frac{\pi 2^{2-d}\Gamma(d-1+k)}{k!(k+(d-1)/2)\Gamma((d-1)/2)^2}\right)^{1/2} \\
&= \sqrt{\omega_d} \mathrm{Lip}(g) \left(O(1) + \sum_{k=1}^{m} O\left(\frac{k}{\Gamma(d-1+k)k!}\right)\right)^{1/2} \\
&= \sqrt{\omega_d} \mathrm{Lip}(g) C(d)
\end{aligned}
$$

for some constant $C(d)$ that only depends on $d$. Hence $\mathrm{Lip}(\hat{g}) = C'(d)\mathrm{Lip}(g)$. $\qquad\square$

The next lemma adapts Lemma 14 to inputs on balls instead of spheres following the construction in (Bach, 2017, Proposition 5).

**Lemma 15.** *Suppose $f : B(0, b) \to \mathbb{R}$ has bounded Sobolev norm $\|f\|_{W^{\nu,2}} \le R$ with $\nu \le (d+2)/2$ even, where $B(0, b) = \{x \in \mathbb{R}^d : \|x\|_2 \le b\}$ is the radius-b ball. Then for every $\epsilon > 0$ there exists $f_\epsilon \in \mathcal{F}_2$ such that $\|f - f_\epsilon\|_{L_2(B(0,b))} = C(d, \nu)b^\nu R\epsilon$, $\mathrm{Lip}(f_\epsilon) \le C'(d)\mathrm{Lip}(f)$, and*

$$\|f_\epsilon\|_{F_2} \le C''(d, \nu)b^\nu R\epsilon^{-\frac{d+3-2\nu}{2\nu}}$$

*Proof.* Define $g(z, a) = f\left(\frac{2bz}{a}\right) a$ on $(z, a) \in \mathbb{S}_d$ with $z \in \mathbb{R}^d$ and $\frac{1}{\sqrt{2}} \le a \in \mathbb{R}$. One may verify that unit-norm $(z, a)$ with $a \ge \frac{1}{\sqrt{2}}$ is sufficient to cover $B(0, b)$ by setting $x = \frac{bz}{a}$ and solve for $(z, a)$. Then we have bounded $\|g\|_{W^{\nu,2}} \le b^\nu R$ and may apply Lemma 14 to get $\hat{g}$ with $\|g - \hat{g}\|_{L_2(\mathbb{S}_d)} \le C(d, \nu)b^\nu R\epsilon$. Letting $f_\epsilon(x) = \hat{g}\left(\frac{ax}{b}, a\right) a^{-1}$ for the corresponding $\left(\frac{ax}{b}, a\right) \in \mathbb{S}_d$ gives the desired upper bounds. $\square$

**Lemma 16.** *Suppose $f : B(0, b) \to \mathbb{R}$ has bounded Sobolev norm $\|f\|_{W^{\nu,2}} \le R$ with $\nu \ge (d+3)/2$ even. Then $f \in \mathcal{F}_2$ and $\|f\|_{F_2} \le C(d, \nu)b^\nu R$.*

*In particular, $W^{\nu,2} \subseteq \mathcal{F}_2$ for $\nu \ge (d+3)/2$ even.*

*Proof.* This lemma reproduces (Bach, 2017, Proposition 5) to functions with bounded Sobolev $L_2$ norm instead of $L_\infty$ norm. The proof follows that of Lemma 14 and Lemma 15 and noticing that by Eq 1,

$$\begin{aligned}
\|g\|_{F_2}^2 &= \sum_{0 \le k : \lambda_k \ne 0} \lambda_k^{-2}\|g_k\|_{L_2(\mathbb{S}_d)}^2 \\
&\le \sum_{0 \le k} k^{d+3-2\nu}\|(\Delta^{\nu/2}g)_k\|_{L_2(\mathbb{S}_d)}^2 \\
&\le \|\Delta^{\nu/2}g\|_{L_2(\mathbb{S}_d)}^2 \\
&\le C_1(d, \nu)\|g\|_{W^{\nu,2}}^2 \\
&\le C_1(d, \nu)R^2.
\end{aligned}$$

$\square$

Finally, we remark that the above lemmas extend straightforward to functions $f : B(0, b) \to \mathbb{R}^{d'}$ with multi-dimensional outputs, where the constants then depend on the output dimension $d'$ too.

### E.3 LEMMA ON APPROXIMATING COMPOSITIONS OF SOBOLEV FUNCTIONS

With the lemmas given above and the fact that the $F_2$-norm upper bounds the $F_1$-norm, we can find infinite-width DNN approximations for compositions of Sobolev functions, which is also pointed out in the proof of Theorem 5.

**Lemma 17.** *Assume the target function $f : \Omega \to \mathbb{R}^{d_{out}}$, with $\Omega \subseteq B(0, b) \subseteq \mathbb{R}^{d_{in}}$, satisfies:*

- *$f = g_k \circ \cdots \circ g_1$ a composition of $k$ Sobolev functions $g_i : \mathbb{R}^{d_i} \to \mathbb{R}^{d_{i+1}}$ with bounded norms $\|g_i\|_{W^{\nu_i,2}}^2 \le R$ for $i = 1, \ldots, k$, with $d_1 = d_{in}$;*

- *$f$ is Lipschitz, i.e. $\mathrm{Lip}(g_i) < \infty$ for $i = 1, \ldots, k$.*

*If $\nu_i \le (d_i + 2)/2$ for any $i$, i.e. less smooth than needed, for depth $L \ge k$ and any $\epsilon > 0$, there is an infinite-width DNN $\tilde{f}$ such that*

- *$\mathrm{Lip}(\tilde{f}) \le C_1 \prod_{i=1}^k \mathrm{Lip}(g_i)$;*

- *$\|\tilde{f} - f\|_{L_2} \le C_2 b^{\frac{\nu_{max}}{2}} R^{\frac{1}{2}}\epsilon$;*

with $\nu_{\max} = \max_{i=1,\dots,k} \nu_i$, the constants $C_1$ depends on all of the input dimensions $d_i$ (to $g_i$) and $d_{out}$, and $C_2$ depends on $d_i, d_{out}, \nu_i, k$, and $\mathrm{Lip}(g_i)$ for all $i$.

If otherwise $\nu_i \geq (d_i + 3)/2$ for all $i$, we can have $\tilde{f} = f$ where each layer has a parameter norm bounded by $C_3 b^{\frac{\nu_{\max}}{2}k} R^{\frac{1}{2}}$, with $C_3$ depending on $d_i, d_{out}$, and $\nu_i$.

*Proof.* Note that by Lipschitzness,

$$(g_i \circ \cdots \circ g_1)(\Omega) \subseteq B\left(0, b \prod_{j=1}^{i} \mathrm{Lip}(g_j)\right),$$

i.e. the pre-image of each component lies in a ball. By Lemma 15, for each $g_i$, if $\nu_i \leq (d_i + 2)/2$, we have an approximation $\hat{g}_i$ on a slightly larger ball $b'_i = b \prod_{j=1}^{i-1} C''(d_j, d_{j+1})\mathrm{Lip}(g_j)$ such that

- $\|g_i - \hat{g}_i\|_{L_2} \leq C(d_i, d_{i+1}, \nu_i)(b'_i)^{\frac{\nu_i}{2}} R^{\frac{1}{2}} \epsilon$;

- $\|\hat{g}_i\|_{F_2} \leq C'(d_i, d_{i+1}, \nu_i)(b'_i)^{\frac{\nu_i}{2}} R^{\frac{1}{2}} \epsilon^{\frac{d_i + 3 - 2\nu_i}{2\nu_i}}$;

- $\mathrm{Lip}(\hat{g}_i) \leq C''(d_i, d_{i+1})\mathrm{Lip}(g_i)$;

where $d_i$ is the input dimension of $g_i$. Write the constants as $C_i, C'_i$, and $C''_i$ for notation simplicity. Note that the Lipschitzness of the approximations $\hat{g}_i$'s guarantees that, when they are composed, $(\hat{g}_{i-1} \circ \cdots \circ \hat{g}_1)(\Omega)$ lies in a ball of radius $b'_i = b \prod_{j=1}^{i-1} C''_j \mathrm{Lip}(g_j)$, hence the approximation error remains bounded while propagating. While each $\hat{g}_i$ is a (infinite-width) layer, for the other $L - k$ layers, we may have identity layers[5].

Let $\tilde{f}$ be the composed DNN of these layers. Then we have

$$\mathrm{Lip}(\tilde{f}) \leq \prod_{i=1}^{k} C''_i \mathrm{Lip}(g_i) = C''(d_1, \dots, d_k, d_{out}) \prod_{i=1}^{k} \mathrm{Lip}(g_i)$$

and approximation error

$$\|\tilde{f} - f\|_{L_2} \leq \sum_{i=1}^{k} C_i (b'_i)^{\frac{\nu_i}{2}} R^{\frac{1}{2}} \epsilon \prod_{j>i} C''_j \mathrm{Lip}(g_j) = O(b^{\frac{\nu_{\max}}{2}k} R^{\frac{1}{2}} \epsilon)$$

where $\nu_{\max} = \max_i \nu_i$, the last equality suppresses the dependence on $d_i, d_{out}, \nu_i, k$, and $\mathrm{Lip}(g_i)$ for $i = 1, \dots, k$.

In particular, by Lemma 16, if $\nu_i \geq (d_i + 3)/2$ for any $i = 1, \dots, k$, we can take $\hat{g}_i = g_i$. If this holds for all $i$, then we can have $\tilde{f} = f$ while each layer has a $F_2$-norm bounded by $O(b^{\frac{\nu_{\max}}{2}k} R^{\frac{1}{2}})$. $\square$

## F   TECHNICAL RESULTS

Here we show a number of technical results regarding the covering number.

First, here is a bound for the covering number of Ellipsoids, which is a simple reformulation of Theorem 2 of (Dumer et al., 2004):

**Theorem 18.** *The $d$-dimensional ellipsoid $E = \{x : x^T K^{-1} x \leq 1\}$ with radii $\sqrt{\lambda_i}$ for $\lambda_i$ the $i$-th eigenvalue of $K$ satisfies $\log \mathcal{N}_2(E, \epsilon) = M_\epsilon(1 + o(1))$ for*

$$M_\epsilon = \sum_{i:\sqrt{\lambda_i} \geq \epsilon} \log \frac{\sqrt{\lambda_i}}{\epsilon}$$

*if one has $\log \frac{\sqrt{\lambda_1}}{\epsilon} = o\left(\frac{M_\epsilon^2}{k_\epsilon \log d}\right)$ for $k_\epsilon = \left|\{i : \sqrt{\lambda_i} \geq \epsilon\}\right|$*

---

[5]Since the domain is always bounded here, one can let the bias translate the domain to the first quadrant and let the weight be the identity matrix, cf. the construction in (Wen & Jacot, 2024, Proposition B.1.3).

For our purpose, we will want to cover a unit ball $B = \{w : \|w\| \leq 1\}$ w.r.t. to a non-isotropic norm $\|w\|_K^2 = w^T K w$, but this is equivalent to covering $E$ with an isotropic norm:

**Corollary 19.** *The covering number of the ball $B = \{w : \|w\| \leq 1\}$ w.r.t. the norm $\|w\|_K^2 = w^T K w$ satisfies $\log \mathcal{N}(B, \|\cdot\|_K, \epsilon) = M_\epsilon (1 + o(1))$ for the same $M_\epsilon$ as in Theorem 18 and under the same condition.*

*Furthermore, $\log \mathcal{N}(B, \|\cdot\|_K, \epsilon) \leq \frac{\mathrm{Tr}K}{2\epsilon^2}(1 + o(1))$ as long as $\log d = o\left(\frac{\sqrt{\mathrm{Tr}K}}{\epsilon}\left(\log \frac{\sqrt{\mathrm{Tr}K}}{\epsilon}\right)^{-1}\right)$.*

*Proof.* If $\tilde{E}$ is an $\epsilon$-covering of $E$ w.r.t. to the $L_2$-norm, then $\tilde{B} = K^{-\frac{1}{2}}\tilde{E}$ is an $\epsilon$-covering of $B$ w.r.t. the norm $\|\cdot\|_K$, because if $w \in B$, then $\sqrt{K}w \in E$ and so there is an $\tilde{x} \in \tilde{E}$ such that $\left\|x - \sqrt{K}w\right\| \leq \epsilon$, but then $\tilde{w} = \sqrt{K}^{-1}x$ covers $w$ since $\|\tilde{w} - w\|_K = \left\|x - \sqrt{K}w\right\|_K \leq \epsilon$.

Since $\lambda_i \leq \frac{\mathrm{Tr}K}{i}$, we have $K \leq \bar{K}$ for $\bar{K}$ the matrix obtained by replacing the $i$-th eigenvalue $\lambda_i$ of $K$ by $\frac{\mathrm{Tr}K}{i}$, and therefore $\mathcal{N}(B, \|\cdot\|_K, \epsilon) \leq \mathcal{N}(B, \|\cdot\|_{\bar{K}}, \epsilon)$ since $\|\cdot\|_K \leq \|\cdot\|_{\bar{K}}$. We now have the approximation $\log \mathcal{N}(B, \|\cdot\|_{\bar{K}}, \epsilon) = \bar{M}_\epsilon(1 + o(1))$ for

$$\bar{M}_\epsilon = \sum_{i=1}^{\bar{k}_\epsilon} \log \frac{\sqrt{\mathrm{Tr}K}}{\sqrt{i}\epsilon}$$

$$\bar{k}_\epsilon = \left\lfloor \frac{\mathrm{Tr}K}{\epsilon^2} \right\rfloor.$$

We now have the simplification

$$\bar{M}_\epsilon = \sum_{i=1}^{k_\epsilon} \log \frac{\sqrt{\mathrm{Tr}K}}{\sqrt{i}\epsilon} = \frac{1}{2}\sum_{i=1}^{\bar{k}_\epsilon} \log \frac{\bar{k}_\epsilon}{i} = \frac{\bar{k}_\epsilon}{2}\left(\int_0^1 \log \frac{1}{x} dx + o(1)\right) = \frac{\bar{k}_\epsilon}{2}(1 + o(1))$$

where the $o(1)$ term vanishes as $\epsilon \searrow 0$. Furthermore, this allows us to check that as long as $\log d = o\left(\frac{\sqrt{\mathrm{Tr}K}}{4\epsilon \log \frac{\sqrt{\mathrm{Tr}K}}{\epsilon}}\right)$, the condition is satisfied

$$\log \frac{\sqrt{\mathrm{Tr}K}}{\epsilon} = o\left(\frac{\bar{k}_\epsilon}{4 \log d}\right) = o\left(\frac{\bar{M}_\epsilon^2}{\bar{k}_\epsilon \log d}\right).$$

$\square$

Second we prove how to obtain the covering number of the convex hull of a function set $\mathcal{F}$:

**Theorem 20.** *Let $\mathcal{F}$ be a set of $B$-uniformly bounded functions, then for all $\epsilon_K = B2^{-K}$*

$$\sqrt{\log \mathcal{N}_2(\mathrm{Conv}\mathcal{F}, 2\epsilon_K)} \leq \sqrt{18}\sum_{k=1}^{K} 2^{K-k}\sqrt{\log \mathcal{N}_2(\mathcal{F}, B2^{-k})}.$$

*Proof.* Define $\epsilon_k = B2^{-k}$ and the corresponding $\epsilon_k$-coverings $\tilde{\mathcal{F}}_k$ (w.r.t. some measure $\pi$). For any $f$, we write $\tilde{f}_k[f]$ for the function $\tilde{f}_k[f] \in \tilde{\mathcal{F}}_k$ that covers $f$. Then for any functions $f$ in $\mathrm{Conv}\mathcal{F}$, we have

$$f = \sum_{i=1}^{m}\beta_i f_i = \sum_{i=1}^{m}\beta_i\left(f_i - \tilde{f}_K[f_i]\right) + \sum_{k=1}^{K}\sum_{i=1}^{m}\beta_i\left(\tilde{f}_k[f_i] - \tilde{f}_{k-1}[f_i]\right) + \tilde{f}_0[f_i].$$

We may assume that $\tilde{f}_0[f_i] = 0$ since the zero function $\epsilon_0$-covers the whole $\mathcal{F}$ since $\epsilon_0 = B$.

We will now use the probabilistic method to show that the sums $\sum_{i=1}^{m}\beta_i\left(\tilde{f}_k[f_i] - \tilde{f}_{k-1}[f_i]\right)$ can be approximated by finite averages. Consider the random functions $\tilde{g}_1^{(k)}, \ldots, \tilde{g}_{m_k}^{(k)}$ sampled iid with

$\mathbb{P}\left[\tilde{g}_j^{(k)} = \left(\tilde{f}_k[f_i] - \tilde{f}_{k-1}[f_i]\right)\right] = \beta_i$. We have $\mathbb{E}[\tilde{g}_j^{(k)}] = \sum_{i=1}^m \beta_i \left(\tilde{f}_k[f_i] - \tilde{f}_{k-1}[f_i]\right)$ and

$$
\begin{aligned}
\mathbb{E}\left\|\sum_{k=1}^K \frac{1}{m_k} \sum_{j=1}^{m_k} \tilde{g}_j^{(k)} - \sum_{k=1}^K \sum_{i=1}^m \beta_i \left(\tilde{f}_k[f_i] - \tilde{f}_{k-1}[f_i]\right)\right\|_\pi^2 &\leq \sum_{k=1}^K \frac{1}{m_k^2} \sum_{j=1}^{m_k} \mathbb{E}\left\|\tilde{g}_j^{(k)}\right\|_\pi^2 \\
&= \sum_{k=1}^K \frac{1}{m_k} \sum_{i=1}^m \beta_i \left\|\tilde{f}_k[f_i] - \tilde{f}_{k-1}[f_i]\right\|_\pi^2 \\
&\leq \sum_{k=1}^K \frac{3^2 \epsilon_k^2}{m_k},
\end{aligned}
$$

where we used the fact that $\left\|\tilde{f}_k[f_i] - \tilde{f}_{k-1}[f_i]\right\|_\pi \leq \epsilon_k + \epsilon_{k-1} = 3\epsilon_k$.

If we choose $m_k = \frac{1}{a_k}\left(\frac{3\epsilon_k}{\epsilon_K}\right)^2$ with $\sum a_k = 1$ we know that there must exist a choice of $\tilde{g}_j^{(k)}$s such that

$$
\left\|\sum_{k=1}^K \frac{1}{m_k} \sum_{j=1}^{m_k} \tilde{g}_j^{(k)} - \sum_{k=1}^K \sum_{i=1}^m \beta_i \left(\tilde{f}_k[f_i] - \tilde{f}_{k-1}[f_i]\right)\right\|_\pi \leq \epsilon_K.
$$

This implies that the finite set $\tilde{\mathcal{C}} = \left\{\sum_{k=1}^K \frac{1}{m_k} \sum_{j=1}^{m_k} \tilde{g}_j^{(k)} : \tilde{g}_j^{(k)} \in \tilde{\mathcal{F}}_k - \tilde{\mathcal{F}}_{k-1}\right\}$ is an $2\epsilon_K$ covering of $\mathcal{C} = \mathrm{Conv}\mathcal{F}$, since we know that for all $f = \sum_{i=1}^m \beta_i f_i$ there are $\tilde{g}_j^{(k)}$ such that

$$
\begin{aligned}
\left\|\sum_{k=1}^K \frac{1}{m_k} \sum_{j=1}^{m_k} \tilde{g}_j^{(k)} - \sum_{i=1}^m \beta_i f_i\right\|_\pi &\leq \left\|\sum_{i=1}^m \beta_i \left(f_i - \tilde{f}_K[f_i]\right)\right\|_\pi \\
&\quad + \sum_{k=1}^K \left\|\frac{1}{m_k} \sum_{j=1}^{m_k} \tilde{g}_j^{(k)} - \sum_{i=1}^m \beta_i \left(\tilde{f}_k[f_i] - \tilde{f}_{k-1}[f_i]\right)\right\|_\pi \\
&\leq 2\epsilon_K.
\end{aligned}
$$

Since $\left|\tilde{\mathcal{C}}\right| = \prod_{k=1}^K \left|\tilde{\mathcal{F}}_k\right|^{m_k} \left|\tilde{\mathcal{F}}_{k-1}\right|^{m_k}$, we have

$$
\begin{aligned}
\log \mathcal{N}_p(\mathrm{Conv}(\mathcal{F}), 2\epsilon_K) &\leq \sum_{k=1}^K \frac{1}{a_k}\left(\frac{3\epsilon_k}{\epsilon_K}\right)^2 \left(\log \mathcal{N}_p(\mathcal{F}, \epsilon_k) + \log \mathcal{N}_p(\mathcal{F}, \epsilon_{k-1})\right) \\
&\leq 18 \sum_{k=1}^K \frac{1}{a_k} 2^{2(K-k)} \log \mathcal{N}_2(\mathcal{F}, \epsilon_k).
\end{aligned}
$$

This is minimized for the choice

$$
a_k = \frac{2^{(K-k)}\sqrt{\log \mathcal{N}_2(\mathcal{F}, \epsilon_k)}}{\sum 2^{(K-k)}\sqrt{\log \mathcal{N}_2(\mathcal{F}, \epsilon_k)}},
$$

which yields the bound

$$
\sqrt{\log \mathcal{N}_p(\mathcal{C}, 2\epsilon_K)} \leq \sqrt{18} \sum_{k=1}^K 2^{K-k}\sqrt{\log \mathcal{N}_2(\mathcal{F}, \epsilon_k)}
$$

$\square$

