# OpenReview forum: "How DNNs break the Curse of Dimensionality: Compositionality and Symmetry Learning"
_ICLR.cc/2025/Conference — ICLR 2025 Poster_

### Official Review · Reviewer_3rVU · 2024-10-27

**Soundness:** 3
**Presentation:** 4
**Contribution:** 3
**Rating:** 6
**Confidence:** 3

**Summary:**

The authors derive generalization bounds for deep networks and functions that are compositions of simpler
functions. The bounds primarily depend on the dimension and regularity of the intermediate layers and functions, which can
potentially have much better properties than the original function. The proof is based on a covering number argument for
a single layer/intermediate function and leverages the fact that covering number behaves well under composition.

**Strengths:**

* The paper is well-written. The presentation is clear and easy-to-follow.
* The covering number argument (proving a covering number result for shallow networks and chaining them) is interesting and
  seems to be very suitable for analyzing deep networks/composition of function. It reduces the problem to a shallow
  one and leads to clean proofs.
* The results are solid and improve on previous bounds when the functions are composition of simpler functions, which
  I believe is a fairly reasonable assumption.

**Weaknesses:**

* My main concern with this paper is about Theorem 4, and I would raise my score if the authors could clarify it. Theorem 4 seems strange to me as it depends only on the largest ratio
  $r_{\max}$ and is independent of the number of layers $L$.
  Given a function $f$, there are multiple ways to decompose
  it into a composition of functions, and it seems possible to reduce the ratio $r_{\max}$ by increasing $L$ and properly
  rescaling each component. Maybe I am missing something here (e.g., there is an $L$-independent lower bound for $r_{\max}$),
  but for now, I do not think the current statement of Theorem 4 is entirely correct. The proof of Theorem 5 (which I
  assume corresponds to Theorem 12 in the appendix) does not resolve my confusion. Specifically, in line 1343, there is a
  summation over $L$ terms that vanishes in the next line.
* Figure 1 (the experiments related to Theorem 2) only shows the scaling laws w.r.t. the size $N$ of the training
  set, while the theoretical results focus more about the dependence on the regularity of the intermediate layers vs
  the global regularity.

* Typos and minor issues:
  * In the statements of most of the results, the failure probability is denoted by $\delta$, but in some bounds it
    is denoted by $p$.
  * The empirical loss is referred to as $\tilde{L}_N$ in some places and by $\hat{L}_N$ in others.
  * Lines 150, 151: the dimensions of $W$ and $V$
  * Line 196: "covering number number"

**Questions:**

* Could you clarify the dependence on $L$ of Theorem 4 (see the Weakness part of the review)?
* I find the remark starting on line 256 unclear. What do you mean by the randomly initialized weights contributing a lot
  to the parameter norm? In practice and most theoretical works, small initialization is used, and the parameter norm
  should not decrease too much, if at all, during training when compared to its initial value.

---

> ### Author Response · Authors · 2024-11-23
>
> Thanks for the thorough review, and for explaining your concerns clearly.
>
> Regarding Theorem 4: The fact that the final rate only depends on the worst ratio is in line with previous results, e.g. (Schmidt-Hieber, 2020). The intuition is that the errors of approximating each function sum up (roughly speaking) therefore only the function with the worst rate matters as $N$ grows. Now things might get a bit more complicated if one considers a number of compositions $L^*$ that grows to infinity. But in this paper we assume $L^*$ to be finite. Also it should not be possible to change $r_{max}$ by adding more layers, because it is determined by the number of times $\nu$ the function can be differentiated, and if $f=g\circ h$ then it is not possible for both $g$ and $h$ to be more times differentiable than $f$ (since the $k$-th derivative of $f$ can be computed from the $1$ to $k$-th derivatives of $g$ and $h$ using Faa di Bruno's formula).
>
> Also thanks for pointing out line 1343 to us, there was indeed an error where we forgot to put the sum over $L$ terms as a constant in front. This constant depends on the $\rho_\ell$s, $r_\ell$s and on the depth $L$ (it scales as $L^{1+r^*}$ in the proof).
>
> - Regarding Figure 1, the focus of our paper is the scaling of generalization bounds $N^{-\gamma}$ as $N$ grows. We felt it made sense to have one plot in terms of $N$ so that one could see that the test error does indeed behave roughly as $N^{-\gamma}$ for some $\gamma$. Figure 2 is the figure that focuses on how $\gamma$ itself is affected by the regularity of the functions $g,h$ that make up the true function $f^* = h \circ g$.
>
> - Thank you for pointing out these typos, we have fixed them.
>
> Regarding your second question: Even though it might be true that one can take small initialization for each individual weight (i.e. a variance of $1/w$ for $w$ the width in LeCun initialization), for large widths $w$ there are more and more parameters. This results in the expected squared parameter norm scaling as $w$, which implies an $F_1$ norm of order $w$, whereas at the end of training the optimal parameter norm should be independent of width.

---

> > ### Comment · Reviewer_3rVU · 2024-11-23
> >
> > Thank you for the clarification. I'll raise my score to 6.

---

### Official Review · Reviewer_4Stc · 2024-11-03

**Soundness:** 4
**Presentation:** 3
**Contribution:** 3
**Rating:** 8
**Confidence:** 3

**Summary:**

The paper aims to proof non-vacuous generalization bounds for deep neural networks by focusing on learning functions that can be decomposed into smooth components (as quantified by Sobolev complexity bounds) that can each be learned by a shallow neural network, all of which are composed to form a deep "accordion" network. They build up to their main result in phases:
* In Theorem 1, they bound the generalization of shallow neural networks on functions with bounded $F_1$ norm and Lipschitz constant.
* They adapt these bounds to deeper networks and compositions of such smooth functions in Theorem 2.
* Proposition 3 replaces the Lipschitzness assumption of the target in Theorem 1 with an bounded-Sobolev norm assumption. They note that this introduces a curse of dimensionality in cases where the Sobolev order is small.
* Theorem 4 analogously extends Proposition 3 to deeper networks, noting that the curse of dimensionality depends on the highest-dimensional component. This implies that the curse of dimensionality can be avoided if the right hierarchy of symmetries exists in the target function such that each layer is low-dimensional.
* Theorem 5 shows that the final loss can be bounded for sufficiently large accordion networks regularized by layer-wise $F_1$ norms that are guaranteed to reach a global minima.

**Strengths:**

As far as I can tell, the mathematics of the paper are correct and this particular decomposition is novel.

I am not particularly concerned about the focus on generalization over optimization and the assumption that the trained neural network attains its global minimum. While this is a substantial assumption, this kind of optimization result would be prohibitively difficult for deep networks, and making this assumption makes it possible to prove interesting results about generalization and decomposition.

**Weaknesses:**

For the sake of clear presentation, I would recommend including examples of the functions that belong to different classes defined herein. For instance, which kinds of functions are Lipschitz but not Sobolev (and vice versa); and what are some examples of targets that can decomposed into a series of Sobolev targets?

## Minor points / nits
* I would add parantheses to the complexity measure on line 70 to make it clear that the product is just for the Lipschitz terms.
* In Theorem 4, it appears that $r^*$ and $r_{\max}$ are confused.

**Questions:**

Is there any particular evidence that can be included about the prevalence of target functions that can be decomposed into low-dimensional components in practical learning settings? It seems to be well-situated within the multi-index model framework for shallow networks, and it's unclear whether compositions of targets like that are thought to be practically relevant.

Is it clear that the $F_1$-norm-based regularization is reasonable for deeper models? If I recall correctly, part of the motivation for the $F_1$ norm in shallow models is that it corresponds with inductive biases for gradient descent, and therefore that bounded $F_1$-norm solutions are likely outcomes even if $F_1$ norm is not explicitly regularized. Are any results like this known for deeper models?

Do these bounds apply when the intermediate function compositions are high-dimensional, but with a low intrinsic dimensionality (e.g. single-index models)?

---

> ### Author Response · Authors · 2024-11-23
>
> Thanks for the positive review.
>
> We have added a note to give an example of a function with a large gap between its $F\_1$ norm and Lipschitz constant "A simple example of a function whose Lipschitz constant is much smaller than its $F_1$ norm (and thus its operator norms since $\\|W \\|\_{op} \geq \frac{\\|W \\|\_F}{\mathrm{Rank} W}$) is is the `zig-zag' function $f(x)=|x - a^{-1}\mathrm{round}(ax)|$ on the interval $[0,1]$: we have $Lip(f)=1$ but $\|f\|_{F_1} = \Omega(a)$ since one neuron is needed for each up or down, each contributing a constant to the parameter norm."
>
> Thanks for pointing out these typos/unclarities, we fixed them.
>
> We already try to give some motivation for why this type of structure might be common (e.g. the presence of symmetries). Another motivation is the section 4.2.2 "Computational Graphs" which suggests that compositional structure might arise on any task that could be described by a computational graph/algorithm, so if we assume that our brain does perform some kind of algorithm to understand text or images or logical problems, then these tasks must have some form of compositional structure. But this is a very fuzzy intuition at the moment, one of our future goal is to formalize this intuition.
>
> Our postulate in this paper. is to view deep networks as a composition of shallow ones, and so if each shallow networks is biased towards minimizing its $F_1$ norm, then the network will be biased towards minimizing the sum of the $F_1$ norm. With weight decay, one is explicitely trying to minimize this sum of the $F_1$ norms, but we hope that in the absence of weight-decay the implicit bias observed in shallow networks would translate to an implicit bias towards small sum of $F_1$ norm. Now the training dynamics of deep networks are really out of reach of theoretical analysis today (outside of the NTK regime of course), so it is difficult to say anything with high confidence. Interestingly, our analysis suggests that it is important to control the Lipschitz constant of each shallow subnetwork too, which seems to not be necessary when working with shallow networks. We cite previous work that suggests that the Lipschitz constant might be controled implicitely by large learning rates, but this is just the beginning.
>
> Yes, we actually already leverage (implicitely) this structure in our proof of theorem 5, where we use the fact that the $F_1$ norms in Theorem 2 can adapt to the inner dimension. This is what allow us to allow the inner dimensions $d\_\ell$ of the network to be larger or equal the true dimensions $d^*_\ell$ (and not exactly match it) without affecting the exponent (though by looking at the proofs one can see that the constant does depend on $d_\ell$ so it is best for $d_\ell$ to not be too large).

---

> > ### Comment · Reviewer_4Stc · 2024-11-23
> >
> > I thank the authors for their detailed response. My score will remain the same.

---

### Official Review · Reviewer_p5Ym · 2024-11-03

**Soundness:** 3
**Presentation:** 3
**Contribution:** 2
**Rating:** 5
**Confidence:** 4

**Summary:**

The paper investigates how deep neural networks (DNNs) can effectively tackle high-dimensional problems by leveraging compositionality and symmetry. Through a novel generalization bound, it demonstrates that DNNs with bounded F1 norms can avoid the curse of dimensionality—something shallow networks struggle with. Using a theoretical framework that combines the F1 norm (a complexity measure) and covering number arguments, the authors show how DNNs can learn compositions of functions by minimizing data requirements. This is particularly achievable when the network’s layers correspond to smooth functions that reduce dimensionality. Some empirical results illustrate phase transitions in learning based on the relative difficulty of learning different functions within the composition, validating the theory's scaling laws.

**Strengths:**

1. The paper seems to be well-thought and of a high-quality in terms of its writing and presentation. Even though the paper is technically involved, I believe that the story is coherent and the mathematical presentation is well-done. For example, the authors use Sec. 1.1 to explicitly explain the type of result they are achieving without too much technical detail, Sec. 2 to provide the necessary information to lay out the framework in which their contribution operates and Sec. 4.1.1 to explain what are the type of results we want to achieve in order to obtain a bound that deals with the curse of dimensionality.
2. As far as I understand the use of the F1-norm (Barron norm) and the per-layer Lipschitz norm to establish generalization bounds is a novel approach in analyzing the generalization capabilities of DNNs. While traditional generalization bounds rely on norms related to the weights, for a given network $f_w$ (without biases), the F1-norm captures the minimal norm of $f_v$ such that $f_v\equiv f_w$. This is a bit original compared to previous work.

**Weaknesses:**

1. Limited comparison and discussion of related bounds. The literature on generalization bounds for neural networks is extensive, with various techniques and approaches, each offering its own angle, contributions, and limitations. I believe the paper lacks sufficient comparisons, either through direct comparisons of the bounds or discussions of the advantages and limitations of other bounds in the literature. For instance, the result in Theorem 2 closely resembles the bounds of Bartlett et al. (2017) and Golowich et al. (2017). While the paper introduces a nice contribution by replacing the spectral norm with the Lipschitz norm in Bartlett 2017 and the Frobenius norm with the F1 norm, it's not clear to me when the new bound is better than the Bartlett bound. Is it possible to directly compare the current bound with those of Bartlett and Golowich?
If so, what are the tradeoffs between these bounds?

Additionally, other works have extended the bounds of Bartlett et al. (2017) and Golowich et al. (2017) to leverage network compositionality for improved guarantees (e.g., Ledent et al. 2019, Graf et al. 2022, Truong 2022, Galanti et al. 2023). How does your bound compare to these works?

Other potentially relevant studies to discuss include those offering relatively tight generalization guarantees on common datasets (e.g., Dziugaite et al. 2017, Zhou et al. 2019, Lotfi et al. 2022).

2. The presentation of the experiments lacks clarity. Much of the experimental setup is confined to the appendix, with certain details—such as the meaning of dashed lines—insufficiently explained. For example, how is the generalization bound in Figure 1 calculated? Does it include constants? When calculating performance, do you take the square root of the loss as in the generalization bound, or not? What is the training loss? How is the $R(\theta)$ term computed? How is the F1-norm determined? The conclusions drawn from the experiments are also unclear. For instance, Figure 1 seems to show that the bound is quite loose and not strongly correlated with test performance.

3. The experiments are conducted in relatively simple settings where the model is a basic fully connected neural network (FCNN) trained on MNIST classification or WESAD. I would expect to see experiments comparable in sophistication to those in the existing literature, where some studies demonstrate that their bounds are relatively tight for MNIST or even more complex datasets, and where the models used are more sophisticated than a basic FCNN (see the references below).

4. While it is common in the literature for many bounds to lack tightness, it would be valuable to see how this bound compares with others in the field. This is particularly important since most bounds in the literature may not be directly comparable to the current bound, and each bound could be tighter in specific settings.

References:

- Zhou et al. 2019: https://openreview.net/pdf?id=BJgqqsAct7

- Dziugaite et al. 2017: http://auai.org/uai2017/proceedings/papers/173.pdf

- Graf et al. 2022: https://proceedings.neurips.cc/paper_files/paper/2022/file/420492060687ca7448398c4c3fa10366-Supplemental-Conference.pdf

- Lotfi et al. 2022: https://openreview.net/pdf?id=o8nYuR8ekFm

- Ledent 2019: https://arxiv.org/abs/1905.12430

- Truong 2022: https://arxiv.org/abs/2208.04284

- Galanti et al. 2023: https://openreview.net/pdf?id=COPzNA10hZ

**Questions:**

1. Are there specific ways to directly compare the current bound with those from previous contributions? For example, how does this bound compare to those of Bartlett et al. (2017), Golowich et al. (2017), etc., for fully connected neural networks (FCNNs), and how does it compare with bounds for compositional architectures such as convolutional neural networks (CNNs)?

2. Would it be possible to evaluate the bound against other bounds in the literature within the authors' proposed learning setting? For instance, could the same models used by the authors be trained, and other bounds calculated for these models to enable comparison?

3. Could the bounds proposed by the authors be applied to more complex architectures, such as convolutional and residual convolutional networks? If so, how would the bounds scale as a function of the number of input/output channels and kernel sizes at each layer? How would this compare with existing studies on bounds for compositional neural networks (e.g., Ledent 2019, Graf et al. 2022)?

4. The authors mention that their bound offers advantages for compositional functions, listing several properties that might contribute to its strength. Is it possible to empirically demonstrate a clear trend where the bound improves as these properties become more pronounced?

---

> ### Author Response · Authors · 2024-11-23
>
> Thanks for your thorough review and the many relevant references.
>
> We agree that the literature on generalization bounds is extensive
> and that we can do a better job of comparing to existing works. But
> these comparison are often difficult since one bound is rarely strictly
> better than another one, we feel like doing too many comparison could
> hurt the readability/accessibility of the paper. We propose to add
> detailed comparison to a few selected results whose comparison is
> easier (e.g. Bartlett et al. 2017), but more importantly we will write
> a high level comparison, which views our work as a combination of
> two lines of work:
>
> On one hand, our work is inspired by approximation results such as
> (Schimidt-Hieber, 2020) which show that compositions of Sobolev/Hölder
> functions can be approximated by neural networks with few parameters,
> so few that the generalization error can be bounded and shown to match
> the known optimal rates for learning such compositions. But these
> results require training networks with few neurons, since they rely
> on a small number of parameters as capacity control, and we know that
> having many neurons helps the optimization of DNNs (e.g. mean-field/NTK
> limits). The work (Galanti et al. 2023, and Poggio et Fraser, 2024)
> can show similar optimal rates under the assumption that sufficiently
> many parameters are zero, which directly constraints the dimensionality
> of the space.
>
> Another approach are norm-based generalization bounds (rather than
> based on dimensionality), which show how the capacity of DNNs can
> be controlled by different complexity measures $R(\theta)$ of the
> parameter that are (almost) independent of the width and depths of
> the network. Many such bounds have been proven in the recent years
> (such as the ones you refer too in your review), but it can be difficult
> to compare them. Also, we are not aware of any result showing that
> these norm-based bounds can achieve tight or close to tight rates
> on compositions of Sobolev functions. To achieve this we make a few
> changes to bound of (Bartlett et al. 2017): switching from operator
> norms to Lipschitz constant, from the (2,1) norm to the $F_{1}$-norm
> (and getting rid of the $2/3$ and $3/2$ exponents, but this has
> a smaller impact). The second change allows us to reuse approximation
> results from (Bach, 2017) to show that compositions of Sobolev functions
> can be approximated by AccNets with almost optimal rates. The first
> change (relying on Lipschitz constant) allows us to retain almost
> optimal rates even in hard settings when multiple functions in the
> compositions are Sobolev with low-differentiability. While it is difficult
> to prove that these changes are necessary to obtain almost optimal
> rates, these two changes allow for a relatively natural construction
> when approximating the composition of Sobolev functions.
>
> In general, we believe that this family of functions could be a good
> baseline to compare generalization bounds, their compositional structure
> seems well-adapted to DNNs, and they contain enough variety to have
> a complex structure to allow for a fair comparison: the complexity
> of the model can either be increased by composing more functions,
> or by increasing the complexity of each function.
>
> We have reworked the introduction to better explain this, and better
> differentiate our work from previous ones (and have also added citations
> to the papers your reference).
>
> We also added a short paragraph after the statement of Theorem 2 comparing
> our bound to the bound of (Bartlett et al. 2017) which is most similar
> to our bound:
>
> "This allows us compare our complexity measure to the
> one in (Bartlett et al. 2017), whose complexity measure $\\prod\_{\\ell}\\|W\_{\\ell}\\|\_{op}\\left(\\sum\_{\\ell}\\frac{\\|W\_{\\ell}\\|\_{2,1}\^{2\/3}}{\\|W_{\\ell}\\|\_{op}\^{2\/3}}\\right)^{3\/2}$
> is closest to ours. We obtain three improvements: replacing the operator
> norm by the Lipschitz constant of $f_{\ell}$, replacing the $(2,1)$-norm
> ($\|A\|\_{2,1}=\|(\|A\_{:,1}\|\_{2},\dots,\|A\_{:,d}\|\_{2})\|\_{1}$) by
> the nuclear norm, and removing the $2/3$ and $3/2$ exponents. These
> three changes play a significant role in Theorem 4, as we discuss
> later. Note however that since our bound relies on the rank of the
> weight matrices being bounded, we cannot say that it is a strict improvement
> over (Bartlett et al. 2017)."
>
> Finally the main change we did is rework the discussion following
> Theorem 4, where we focus on 4 points: the power of already existing
> approximation results for the $F_{1}$-norm; sparsity-based bounds
> vs our norm based bounds (comparison to (Schimidt-Hieber, 2020) and others);
> how the Lipschitz constant vs operator norms affect the final rates;
> the large depth behavior of our bound in comparison to (Bartlett et
> al. 2017) and (Golowich et al. 2020) which can in some case lead to
> an exponential dependence on depth. You can check the updated version for details of this discussion.

---

> > ### Author Response · Authors · 2024-11-23
> >
> > Regarding your questions:
> > 1) We have added detailed comparisons to  (Bartlett et al. 2017) and (Schmidt-Hieber, 2020) since these are the most similar bounds. While our bound could be adapted to work with CNNs, it is not obvious at this stage how exactly it would look like (and CNNs are not particularly adapted to the composition of Sobolev functions we consider). For this reason we only mention the CNN generalization bounds that you pointed to, since comparison is not very meaningful.
> >
> > 2) We are currently running numerical experiments to compare with previous bounds. The difficulty is that the Lipschitz constant cannot be evaluated, we therefore will compute two values: the upper bound in term of the operators (which yields a similar final bound as (Bartlett et al. 2017)) and a lower bound by taking the max of the Jacobian operator norm over random points (which is somewhat similar to (Wei \& Ma,
> > 2019)). We can see a significant gap in our preliminary results, suggesting that the change to the Lipschitz constant might the most impactful improvement.
> >
> > 3) We are confident that could be generalized to CNNs, but this was not the focus of this paper. The focus of this paper was to show that one can use norm-based bounds to give learning guarantees for compositional functions.
> >
> > 4) It is non-trivial to prove that it is impossible to use previous bounds to prove an analogue to Theorem 5, but the second part of Theorem suggests that without at least switching to Lipschitz constants or an analogue it might not be possible to obtain tight compositional rates. This is a setting  where the compositionality can lead to an infinitely better bound.

---

> > > ### Comment · Reviewer_p5Ym · 2024-11-26
> > > **Thank you for the additional discussions and clarifications**
> > >
> > > I appreciate the authors additional discussions of other works in the field, the specific comparison with Bartlett's bound and the clarifications of their contribution. I am raising my score to 5. I am still curious about the direct empirical comparisons with previous bounds in the literature. Depending on the results, I am happy to increase the score even further.

---

> > > > ### Author Response · Authors · 2024-11-27
> > > >
> > > > Thank you for raising your score. We have just finished our experiments to compare to other bounds, and we do see a significant improvement from our bound over previous ones, and this difference is not just a result of switching from operator norm to Lipschitz constnat, since our approximate bound with the operator norm also yields an improvement. For a more detailed discussion, you can check Figure 4 in the appendix of the updated version.

---

### Official Review · Reviewer_od6v · 2024-11-04

**Soundness:** 3
**Presentation:** 3
**Contribution:** 3
**Rating:** 6
**Confidence:** 2

**Summary:**

The paper explores how deep neural networks (DNNs) can overcome the curse of dimensionality through mechanisms of compositionality and symmetry learning. It introduces Accordion Networks (AccNets), which are structured as compositions of shallow subnetworks, and derives new generalization bounds based on the F1-norm and Lipschitz properties of these subnetworks. The authors prove that, under certain conditions, DNNs can efficiently learn complex functions, such as compositions of Sobolev functions, by leveraging these structural properties. Empirical experiments support the theoretical results, demonstrating phase transitions in learning behaviors and illustrating how compositionality helps avoid the exponential data requirements that typically plague high-dimensional learning tasks.

**Strengths:**

- The paper introduces an innovative framework to explain how DNNs overcome the curse of dimensionality through compositionality and symmetry learning. Using Accordion Networks (AccNets) to model compositions of functions and deriving generalization bounds based on F1-norms and Lipschitz constants is a novel and insightful contribution.
- The theoretical analysis is thorough, with detailed proofs and a solid foundation in functional analysis. The empirical results, though limited in scope, align well with the theory, demonstrating meaningful phase transitions and supporting the derived bounds.
- The paper is well-organized and logically presented, though the advanced mathematical concepts may still be challenging for non-expert readers. Key ideas are well-articulated, but additional intuition would improve accessibility.
- The paper has the potential to make a significant impact on our understanding of deep learning's success. By framing generalization in terms of compositionality and symmetry, the work could inspire new approaches to model design and optimization.

**Weaknesses:**

- The experiments are limited to small-scale models, and the applicability of the results to large-scale, real-world networks remains unclear. More extensive testing on modern architectures would strengthen the findings.
- The reliance on smoothness and regularity assumptions (e.g., Sobolev norms) may limit the practical relevance of the results. The paper does not sufficiently address how deviations from these assumptions affect performance.
- The work lacks clear guidelines for implementing the theoretical insights in real-world scenarios. The two regularization methods proposed are difficult to optimize and may not be straightforward to apply.

**Questions:**

- How sensitive are the theoretical results to violations of the regularity assumptions?
- Can the authors provide clearer guidelines for when to use each regularization method?
- Are there alternative methods to control Lipschitz constants that could simplify optimization?

---

> ### Author Response · Authors · 2024-11-23
>
> Thank you for the thoughtful review and interesting questions.
>
> Regarding the weaknesses:
>
> - Our experiments were designed to illustrate the results, so we chose
> a family of tasks where the compositionality can be tuned to reveal
> the phase changes and other predictions of our theory. We agree that
> it would be interesting to test our theory on larger text or image
> datasets, though our intuition is that we would first have to extend
> our theory to CNNs/transformers to meaningfully train on such tasks.
>
> - Actually the point of the composition of Sobolev functions structure
> is that much less regularity is required for efficient learning: the
> true function could be the composition of a first smooth function
> that maps to a one dimensional feature space followed by a function
> that is only Lipschitz; such a composition can have non-differentiable
> points (if the second function is non-differentiable) and still be
> learnable efficiently. Without compositionality, the only learnable
> functions are those that are roughly as many times differentiable
> as the input dimension.
>
> - That's a very good point, we should make clearer recommendations.
> However our recommendation is to train DNNs as usual (using traditional
> weight decay/L2 regularization) and this what we do in our experiments.
> As we explain at the lines 458-466, we believe that
> the use of large learning rates implicitly controls the Lipschitzness,
> so only the F1 norms need to be minimized using weight decay. The
> problem is that at the moment, there is only partial theoretical support
> for this implicit control, so it is difficult push for this choice.
>
> We plan to study this implicit control in more details and hopefully
> prove it in future work. The intuition is simple: if the functions
> that map from some hidden representations to the outputs are not Lipschitz,
> then the gradients will blow up, leading to unstability under large
> learning rates. This control can easily be proven in Linear networks,
> so we could add this in the appendix as further proof.
>
> Nevertheless we have reworked the above mentioned paragraph to more
> explicitely advocate for the use of L2 regularization together with
> large learning rates.
>
> Regarding your questions:
>
> - When you say regularity assumption, do you mean Lipschitzness or
> the Sobolev norms? The Sobolev norms are not strictly necessary, they
> are just a common functional norm that can be used to bound the F1
> norm, one could instead work with functions with bounded F1 norms
> directly (i.e. Theorem 2). For the Lipschitzness assumption, it seems
> pretty clear that one needs to control how errors in the middle of
> the network propagate to the outputs, but something weaker than Lipschitzness
> might be sufficient (e.g. a bound on the Jacobian on the training
> data only, as in (Wei \& Ma, 2019) which can be computed), we hope
> to investigate these ideas in the future.
>
> - See our response to the third weakness.
>
> - We mention above the implicit bias of large learning rates, or weakening
> Lipschitzness to a maximum over the training data as in (Wei \& Ma,
> 2019).

---

> > ### Comment · Reviewer_od6v · 2024-11-27
> >
> > I appreciate the detailed response. I maintain my positive rating.

---

### Author Response · Authors · 2024-11-23
**Revision**

We have made a number of changes, mainly to better compare our result to previous generalization bounds, and better explain the effect of the Lipschitz constant and depths. The bigger changes are highlighted in red.

We are also running experiments to compare our bounds to previous ones and will update the paper again before the end of the discussion period.

---

> ### Author Response · Authors · 2024-11-27
>
> We have made a final change, adding a figure which compares our bound to previous bound. We see a significant improvement, though it still remains far from being tight. The new figure is in the appendix, Figure 4.

---

### Meta-Review · Area_Chair_sb5M · 2024-12-20

**Metareview:**

This paper investigates how deep neural networks (DNNs) can avoid the curse of dimensionality when learning compositions of functions. The authors derive generalization bounds based on the F1 norm and covering number arguments, demonstrating that DNNs can efficiently learn complex functions if they are composed of simpler components.

The primary weaknesses include limited experimental validation, reliance on potentially restrictive smoothness assumptions, and lack of practical guidelines for implementing the proposed regularization methods. The work's strengths lie in its novel theoretical framework, thorough mathematical analysis, and potential to significantly impact the understanding of deep learning's success. This paper presents a strong theoretical contribution to understanding DNN generalization, with clear potential for inspiring new research directions. Therefore, I recommend acceptance.

**Additional Comments On Reviewer Discussion:**

The reviewers raised several concerns regarding the practical applicability and empirical validation of the proposed generalization bounds. The authors responded by clarifying that their experiments were designed to illustrate theoretical predictions rather than test on large-scale datasets, and that they intend to extend their theory to CNNs/transformers in future work. They further addressed concerns about the reliance on smoothness assumptions by arguing that their framework actually requires less regularity than traditional approaches due to its compositional structure.

Regarding comparisons with existing generalization bounds, the authors acknowledged the need for a more thorough discussion and added detailed comparisons to a few key results, highlighting the improvements offered by their bound, especially in settings with compositions of low-regularity functions. The authors also added new experimental results and a figure. They also clarified that their focus was on demonstrating that norm-based bounds can provide learning guarantees for compositional functions and acknowledged the difficulty of proving that existing bounds could not achieve similar results.

Overall, it seems most of the main concerns were addressed in the rebuttal.

---

### Decision · Program_Chairs · 2025-01-22

Accept (Poster)